# DotMatch: Simplified Semi-Supervised Learning with the Log Dot Product Loss

## Abstract

Semi-supervised learning (SSL) algorithms typically work by generating supervisory signals for unsupervised data using the model being trained, but such supervisory signals are generally imperfect, thus various techniques have been proposed to balance the signal-to-noise ratio, such as confidence-based pseudo-labeling, consistency regularization and entropy regularization. However, these methods often require careful tuning of hyperparameters, such as the confidence threshold in pseudo-labeling and the regularization strength in regularization methods, which is often a challenging task, particularly with limited labeled data available for validation. In this paper, we introduce DotMatch, an SSL algorithm that is capable of balancing the signal-to-noise ratio without any algorithm specific hyperparameters. Specifically, we introduce a novel consistency loss on unsupervised data to replace the cross-entropy loss, called the log dot product (LDP) loss, which is simply the negative log of the dot product between the predicted label distributions of weak and strong augmented views of an input. We empirically and theoretically demonstrate that the LDP loss enjoys several benefits for SSL, compared to the cross-entropy loss with soft target: nonconfident examples have low impacts on model updates, as in confidence-based pseudo-labeling methods such as SoftMatch; predictions are encouraged to have a low entropy, as in entropy-regularized methods; and interestingly, its gradient is appropriately scaled relative to the gradient of the supervised loss, thus requiring no regularization constant. We additionally combine the LDP loss with distribution alignment to ensure the distribution of predictions on unlabeled data match that of the labeled data. Extensive experiments show that DotMatch is competitive with state-of-the-art baselines without needing to tune any algorithm-specific hyperparameters for different datasets. Code is available at: [open source upon acceptance].

## 1 Introduction

Deep learning has seen great success in many supervised learning problems (He et al., 2016; Vaswani et al., 2017; Dong et al., 2018), but requires lots of labeled training data which is often difficult to obtain. Semi-supervised learning (SSL) (Zhu, 2005; Zhu & Goldberg, 2009) eases this burden by enabling the use of supplemental unlabeled data which is much easier to obtain, for example, collect lots of images without manually annotating them. Many SSL algorithms use a form of self-training (McLachlan, 1975; Rosenberg et al., 2005; Lee et al., 2013; Xie et al., 2020b) where, during training, supervision for the unlabeled data is provided by the model itself. Since the model is only capable of providing inaccurate supervision, the goal is to maintain a high signal-to-noise ratio whereby we get abundant highly accurate supervision with minimal noise (similar to the quantity-quality tradeoff (Chen et al., 2023)).

Several techniques and their combinations are often utilized in SSL algorithms, such as (a) pseudo-labeling (Lee et al., 2013; Sohn et al., 2020), which generates labels for unlabeled data using the current model, then uses such pseudo-labels to further train the model; (b) consistency regularization (Bachman et al., 2014; Sajjadi et al., 2016b; Laine & Aila, 2017) which uses the model's prediction after randomly perturbing the input or the model itself as a supervision signal; (c) entropy minimization (Grandvalet & Bengio, 2004a), which ensures that the model learns to make confident predictions on unlabeled data; (d) distribution alignment (Bridle et al., 1991; Berthelot et al., 2020), which encourages the model to match the average predicted label distribution on unlabeled examples with the distribution of observed labels. However, these methods often require

sophisticated balancing of signal-to-noise ratio. For example, many state-of-the-art SSL algorithms perform pseudo-labeling and propose carefully crafted pseudo-label confidence threshold schemes to improve performance (Zhang et al., 2021; Wang et al., 2023; Chen et al., 2023; Li et al., 2023) rather than using a constant threshold (Sohn et al., 2020). In addition, existing methods often require careful tuning of hyperparameters, such as the regularization strength in regularization methods.

We propose a SSL algorithm, DotMatch, that provides a simple approach to control the signal-to-noise ratio, and avoids the challenging task of tuning algorithm-specific hyperparameters as it does not have such hyperparameters. DotMatch uses a new loss function, called the log dot product (LDP) loss, which serves as an alternative generalization of the CE loss with a single label to soft weighted labels. Instead of considering the dot product between the soft target distribution and the logarithm of the predicted label distribution as is done with the CE loss, the LDP loss instead computes the logarithm of the dot product between the two distributions. As we shall demonstrate empirically and theoretically in Section 4, this approach has several advantages in the context of SSL: low-confidence examples naturally contribute less to the gradient update without requiring additional confidence-based loss weights as is typically used with pseudo-labeling (Sohn et al., 2020); the model will automatically learn to produce high confidence predictions (Grandvalet & Bengio, 2004a; Sajjadi et al., 2016a) without needing to rely on inaccurate one-hot pseudo-labels, additional entropy regularization which requires tuning of the regularization strength, or target sharpening which requires tuning the sharping hyperparameter; and the gradients are appropriately scaled relative to the gradients of the supervised CE loss, thus no additional regularization constant is necessary. We combine the LDP loss with advanced data augmentation techniques (Xie et al., 2020a) in DotMatch to encourage the model to make the same predictions for different augmented versions of the same input. Finally, distribution alignment (Berthelot et al., 2020) is applied to the targets in the LDP consistency regularizer to ensure that the average predicted label distribution on unlabeled examples matches that of the labeled data, and does not collapse into a single class in order to achieve high confidence and consistent predictions. DotMatch demonstrates strong performance in extensive experiments. Notably, this is achieved without the need to tune algorithm specific hyperparameters such as confidence threshold or regularization strength, allowing it to perform well on new problems without requiring extra labeled data for validation.

Our contributions are as follows:

- We provide a motivating example to show explicitly that down-weighting the loss on low-confidence unlabeled examples leads to improved gradient quality in SSL, and in turn improves the predictive performance of learned models.

- We introduce the novel log dot product loss for consistency regularization which naturally reduces the influence of low-confidence examples on the model updates without requiring additional confidence-based loss weights, encourages the model the produce confident predictions on unlabeled data without introducing any hyperparameters that require tuning, and does not require any regularization constant due to the scale of the gradient matching that of the supervised CE loss.

- We further propose an SSL algorithm, DotMatch, which uses the LDP loss for consistency regularization combined with distribution alignment to encourage the distribution of predictions on unlabeled examples to match that of the labeled examples.

- A novel theoretical analysis of SSL consistency loss functions is performed to explain their efficacy, and explore the relationship between loss gradients and confidence.

- Extensive experiments on SSL benchmark problems demonstrate that DotMatch is competitive with state-of-the-art baselines despite its simplicity. The ablation study shows that the components of DotMatch positively contribute to performance.

## 2 RELATED WORK

Our work is closely related to many ideas and algorithms from the SSL literature. We present some of these in this section, with broader introductions to SSL available in, for example, (Zhu, 2005; van Engelen & Hoos, 2020).

Consistency regularization (Bachman et al., 2014; Sajjadi et al., 2016b; Laine & Aila, 2017) is commonly used in SSL algorithms as a natural way to incorporate unlabeled data into the training

process of a supervised learning model. At a high level, the goal is to encourage the model to produce similar outputs for similar inputs. One way to achieve this is by training the model to make the same prediction on different augmented versions of the same input. This approach is particularly effective when using high quality data augmentation strategies (Xie et al., 2020a). Specifically, the model is encouraged to match the prediction on a strongly augmented input with that of a weakly augmented version of the same input. Some SSL methods have proposed alternative measures of consistency, such as encouraging consistency between different versions of the model (Chen et al., 2022; Huang et al., 2023), unlabeled and labeled datasets (Huang et al., 2024; Heidari et al., 2024), a subset of the class predictions (Wu & Cui, 2024; Yang et al., 2023), examples from the same cluster (Liu et al., 2025), adding inconsistency regularization between different examples (Deng et al., 2024), or using ideas from optimal transport to measure consistency (Tan et al., 2024). DotMatch continues the work along this direction by proposing a new consistency loss.

Entropy regularization (Grandvalet & Bengio, 2004a) is another important idea in SSL that encourages the model to produce high-confidence predictions on unlabeled examples. Combining entropy regularization with consistency regularization ensures that the model does not converge to undesirable local minima where perfect consistency may be achieved by predicting labels for all examples uniformly at random. Target sharpening (Berthelot et al., 2019; Xie et al., 2020a; Berthelot et al., 2020) is one method that has been proposed to incorporate entropy minimization into SSL. Another technique is pseudo-labeling (Lee et al., 2013) with one-hot pseudo-labels. Since pseudo-labels generated by the model are generally inaccurate, especially early in training, it is common to only utilize pseudo-labels that the model is confident about. Some confidence threshold schemes include using a constant threshold (Sohn et al., 2020), dynamic thresholds which either depend on the pseudo-label class (Zhang et al., 2021; Wang et al., 2023) or not (Chen et al., 2023), thresholding an alternative measure of confidence (Min et al., 2024), or using auxiliary models to measure confidence or provide thresholds (Li et al., 2023; 2024; Fang et al., 2024). Since the LDP loss naturally minimizes the entropy of predictions and focuses more on highly confident examples, we do not use any target sharpening, one hot pseudo-labeling, confidence thresholding or auxiliary models in DotMatch.

Distribution Alignment (DA) (Berthelot et al., 2020; 2022) modifies the pseudo-labels such that the average pseudo-label distribution aligns better with the labeled data label distribution. This effect can also be achieved with loss regularization (Bridle et al., 1991; Wang et al., 2023). DA is particularly beneficial when the number of classes is large (Sohn et al., 2020), as the model predictions are susceptible to collapse into a single class in order to achieve consistent and low entropy predictions. We use the former type of DA in DotMatch as it does not introduce any additional regularization hyperparameters that require tuning.

Objective functions similar to the LDP-based consistency loss were proposed in (Hsu et al., 2019; Cao et al., 2022), except different problem settings were considered in both works so the purpose of the objectives is different, they did not incorporate weak-strong augmentations for consistency or DA, and we provide additional insights into the properties of the loss through an extensive analysis of the gradients.

## 3 PROBLEM SETTING

Let $\boldsymbol{x} \in \mathbb{R}^d$ denote an input and $\boldsymbol{y} \in \{0,1\}^K \cap \Delta$ be a one-hot class label, where $\Delta = \{\boldsymbol{q} \in [0,1]^K : \boldsymbol{q}^\top \mathbf{1} = 1\}$ is the standard $K-1$ simplex. Following standard practice (Zhang et al., 2021; Wang et al., 2022), we consider the setting where data comes in mini-batches of labeled and unlabeled examples having data augmentation applied. In each training iteration we receive a batch of labeled examples $\mathcal{L} = \{(\boldsymbol{w}(\boldsymbol{x}_i), \boldsymbol{y}_i)\}_{i=1}^{n_L}$ with each $(\boldsymbol{x}_i, \boldsymbol{y}_i)$ distributed according to some unknown joint distribution $p(\boldsymbol{x}, \boldsymbol{y})$, and a batch of unlabeled examples $\mathcal{U} = \{(\boldsymbol{w}(\boldsymbol{x}_i), \boldsymbol{s}(\boldsymbol{x}_i))\}_{i=1}^{n_U}$ with each $\boldsymbol{x}_i$ being drawn from the marginal distribution over inputs $p(\boldsymbol{x}) = \sum_{\boldsymbol{y}} p(\boldsymbol{x}, \boldsymbol{y})$. The weak ($\boldsymbol{w}$) and strong ($\boldsymbol{s}$) augmentation functions $\boldsymbol{w}, \boldsymbol{s} : \mathbb{R}^d \to \mathbb{R}^d$ are stochastic. Some example weak augmentations are random crop, flip, shift, and strong augmentations include CutOut (DeVries & Taylor, 2017), CTAugmnet (Berthelot et al., 2020), RandAugment (Cubuk et al., 2020), or data modality agnostic augmentations such as MixUp (Zhang et al., 2018) or adversarial perturbations (Miyato et al., 2019). For the model we use a neural network $\boldsymbol{p}(\cdot\,; \boldsymbol{\theta}) : \mathbb{R}^d \to \Delta$ with softmax output layer, and learnable parameters collected in the vector $\boldsymbol{\theta}$. We consider SSL objective functions of

the form

$$L(\boldsymbol{\theta}) = \frac{1}{|\mathcal{L}|} \sum_{(\boldsymbol{w}, \boldsymbol{y}) \in \mathcal{L}} \ell_l(\boldsymbol{p}(\boldsymbol{w}; \boldsymbol{\theta}), \boldsymbol{y}) + \frac{\lambda}{|\mathcal{U}|} \sum_{(\boldsymbol{w}, \boldsymbol{s}) \in \mathcal{U}} w(\max(\boldsymbol{p_w})) \ell_u(\boldsymbol{p}(\boldsymbol{s}; \boldsymbol{\theta}), \boldsymbol{p_w}) \qquad (1)$$

for some classification loss functions $\ell_l, \ell_u : \Delta \times \Delta \to \mathbb{R}_{\geq 0}$, and non-decreasing weight function $w : [1/K, 1] \to [0, 1]$. We abuse the notations $\boldsymbol{w}$ and $\boldsymbol{s}$ to denote a weak and a strong augmented example respectively, and we use $\boldsymbol{p_x} = \boldsymbol{p}(\boldsymbol{x}; \boldsymbol{\theta})$ to denote the prediction on example $\boldsymbol{x}$, which is not used in gradient computation. The unlabeled loss weight $\lambda$ is typically set to 1. Many existing SSL algorithms have objectives of this form, for example, (Sohn et al., 2020; Zhang et al., 2021; Chen et al., 2023; Wang et al., 2023). At test time, the predicted label for new example $\boldsymbol{x} \in \mathbb{R}^d$ is given by $\operatorname{argmax}(\boldsymbol{p_x})$. Throughout the paper we use $\boldsymbol{e}_i \in \{0, 1\}^K \cap \Delta$ to denote the one-hot label for class $i$.

## 4 DOTMATCH

In this section we first use a motivating example to highlight an important SSL algorithm design consideration: down-weighting low-confidence unlabeled examples in the loss, which leads to improved gradient quality and, in turn, improved predictive performance. We then introduce our new LDP loss function and SSL algorithm, DotMatch, and give theoretical justification for their efficacy.

### 4.1 A MOTIVATING EXAMPLE

A desirable quality of an SSL algorithm is to have unlabeled examples with low-confidence pseudo-labels down-weighted in the computation of the loss to reduce the impact of confirmation bias during training, since higher confidence pseudo-labels tend to be correct more often than low-confidence ones. This property can equivalently be viewed through the lens of gradient-based learning: we want the contribution to the gradient from low-confidence examples to be smaller than that of high-confidence examples, so that the model update will be less influenced by low-confidence examples. Many recent SSL algorithms that weight the loss for each unlabeled example based on the prediction confidence have this property.

We compare one representative algorithm, SoftMatch (Chen et al., 2023), against a simple SSL algorithm that replaces the SoftMatch weighting function with the constant function $w(\alpha) = 1$ for each unlabeled example, which we refer to as CE(hard), since $\ell_u$ in this case is the CE loss with hard one-hot pseudo-label. We train each algorithm separately on EMNIST (Cohen et al., 2017) with 4 labeled examples per class. For a detailed description of the experiment settings, we kindly refer you to Section 5. After 5000 training iterations, before training is finished, we record the target confidence and gradient norm

$$(\max(\boldsymbol{p_w}), \|\nabla_{\boldsymbol{\theta}} \, w(\max(\boldsymbol{p_w})) \ell_u(\boldsymbol{p}(\boldsymbol{s}; \boldsymbol{\theta}), \boldsymbol{p_w})\|_2)$$

for each unlabeled example in the batch. We also look at the quality of the gradients associated with unlabeled data as measured by the proportion of total gradient norm coming from correctly pseudo-labeled examples

$$\text{Quality}(w, \ell_u) = \frac{\frac{1}{|\mathcal{U}|} \sum_{(\boldsymbol{w}, \boldsymbol{s}, \boldsymbol{y}) \in \mathcal{U}} \mathbb{1}(\boldsymbol{y} = \boldsymbol{e}_{\operatorname{argmax}(\boldsymbol{p_w})}) \|\nabla_{\boldsymbol{\theta}} \, w(\max(\boldsymbol{p_w})) \ell_u(\boldsymbol{p}(\boldsymbol{s}; \boldsymbol{\theta}), \boldsymbol{p_w})\|_2}{\frac{1}{|\mathcal{U}|} \sum_{(\boldsymbol{w}, \boldsymbol{s}, \boldsymbol{y}) \in \mathcal{U}} \|\nabla_{\boldsymbol{\theta}} \, w(\max(\boldsymbol{p_w})) \ell_u(\boldsymbol{p}(\boldsymbol{s}; \boldsymbol{\theta}), \boldsymbol{p_w})\|_2}.$$

$$(2)$$

We further report the test accuracy after training is completed at 50 000 iterations. The quality measure in Equation (2) is similar in spirit to the quality of pseudo-labels from (Chen et al., 2023), but applied to gradient norms. Note that the gradient quality is not computable in general SSL problems due to the use of ground truth labels for the unlabeled examples, and is only used here for illustrative purposes. Results are reported in Figure 1.

We learn a few things from the results. Firstly, incorrect pseudo-labels tend to have lower confidence than correct ones for both methods. SoftMatch reduces the influence of incorrectly pseudo-labeled examples on the gradient by explicitly down-weighting the contribution of low-confidence pseudo-labels, whereas CE(hard) has no mechanism for doing this and simply treats all unlabeled examples equally. This results in SoftMatch having higher quality gradients compared to CE(hard). Overall, down-weighting the gradient norm of low-confidence unlabeled examples reduces confirmation bias from incorrect pseudo-labels during training, in turn improving performance of the final model.

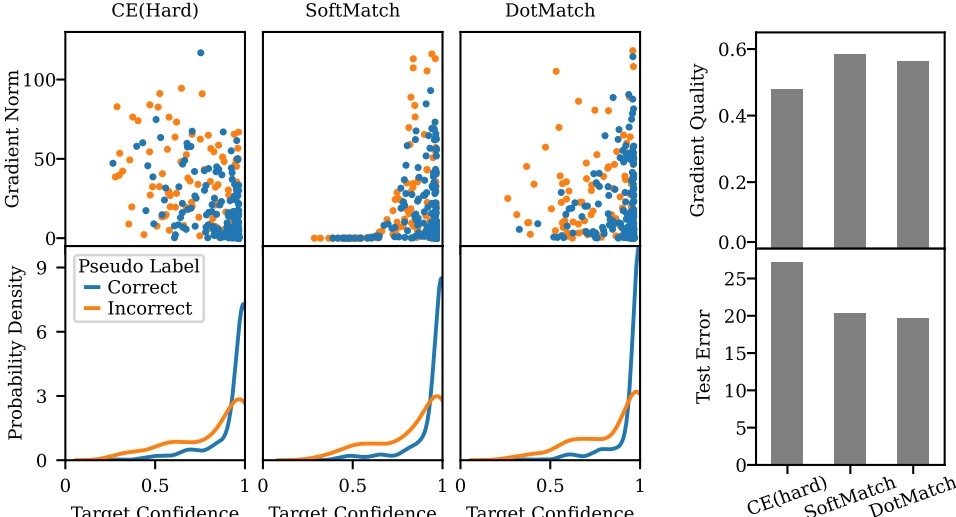

Figure 1: Comparison of the unlabeled example gradients and performance of CE(hard), Soft-Match, and our algorithm DotMatch, which is motivated by the differences between CE(hard) and SoftMatch. Left: Gradient norm (top) and density (bottom) vs confidence. CE(hard) has most large-gradient unconfident examples, SoftMatch least, and DotMatch in-between. Right: Gradient quality (top) and test error (bottom). SoftMatch and DotMatch both higher gradient quality and lower test error compared to CE(hard). Results recorded after 5000 training iterations on EMNIST (Cohen et al., 2017) with 4 labels per class, except test error, which was recorded after completing 50k training iterations.

The LDP loss used in our DotMatch algorithm is motivated by the above observation on SoftMatch's ability to down-weight low-confidence examples in learning, and Figure 1 demonstrates DotMatch's ability to do so too. In the next section we introduce the LDP loss and provides some theoretical insights on its ability to down-weight the gradient norm of low-confidence examples, which in turn produces high quality gradients and better performance of the final learned model.

### 4.2 LOG DOT PRODUCT LOSS

The de facto unlabeled loss function $l_u$ is the CE loss with a single pseudo-label generated from the model. However, this ignores the full information about the predicted label distribution. One way to generalize the CE loss with a single label $\ell(\boldsymbol{q}, \boldsymbol{e}_y) = -\log q_y$ for some fixed $y \in \{1, \ldots, K\}, \boldsymbol{q} \in \Delta$, to multiple weighted labels $\boldsymbol{p} \in \Delta$ is $\ell(\boldsymbol{q}, \boldsymbol{p}) = -\boldsymbol{p}^\top \log \boldsymbol{q}$. We refer to this loss as CE(soft). However, it is not the only generalization. Another option is to look at the negative log of the dot product between distributions $\ell(\boldsymbol{q}, \boldsymbol{p}) = -\log(\boldsymbol{p}^\top \boldsymbol{q})$. We refer to this loss function as the log dot product loss. This loss has many interesting properties that we highlight below.

**Consistency and Entropy Minimization** A prediction $\boldsymbol{q}$ that minimizes the LDP loss with target $\boldsymbol{p}$ is one that puts all of its probability mass on the set of maximizers of the target

$$\operatorname*{argmin}_{\boldsymbol{q} \in \Delta} -\log(\boldsymbol{p}^\top \boldsymbol{q}) = \{\boldsymbol{q} \in \Delta \ : \ q_k = 0 \, \forall \, k \notin \operatorname{argmax}(\boldsymbol{p})\}.$$

This occurs when the dot product inside the logarithm achieves its maximum value. In the common situation where there is a unique maximizer in $\boldsymbol{p}$, the argmin is the singleton containing $\boldsymbol{e}_{\operatorname{argmax}(\boldsymbol{p})}$. In this common situation, the LDP loss is minimized when the target and prediction are both equal and have zero entropy. Compare with CE(soft) loss:

$$\operatorname*{argmin}_{\boldsymbol{q} \in \Delta} -\boldsymbol{p}^\top \log(\boldsymbol{q}) = \boldsymbol{p},$$

which only encourages low entropy predictions if the target $\boldsymbol{p}$ has low entropy (for example, one-hot pseudo-label). LDP loss has an entropy minimization effect without the need for one-hot targets, temperature scaled softmax targets, or additional entropy regularization loss term.

Table 1: Minimizers and gradients of different loss functions, where $\boldsymbol{p}$ denotes the target distribution, $\widehat{y}$ the index of $\boldsymbol{p}$ with largest value, $\boldsymbol{z}$ the logits, $\boldsymbol{\theta}$ the model parameters, $\boldsymbol{q}$ the predicted distribution (either as a function of $\boldsymbol{z}$ or $\boldsymbol{\theta}$), and $\odot$ the element-wise product. Note that $\boldsymbol{q} = \mathrm{softmax}(\boldsymbol{z})$.

| Name | $\ell(\boldsymbol{q}, \boldsymbol{p})$ | $\mathrm{argmin}_{\boldsymbol{q}}\ell(\boldsymbol{q}, \boldsymbol{p})$ | $\nabla_{\boldsymbol{z}}\ell(\boldsymbol{q}(\boldsymbol{z}), \boldsymbol{p})$ | $\nabla_{\boldsymbol{\theta}}\ell(\boldsymbol{q}(\boldsymbol{\theta}), \boldsymbol{p})$ |
|---|---|---|---|---|
| CE(soft) | $-\boldsymbol{p}^{\top}\log\boldsymbol{q}$ | $\boldsymbol{p}$ | $\boldsymbol{q} - \boldsymbol{p}$ | $-\sum_{k=1}^{K}\frac{p_k}{q_k(\boldsymbol{\theta})}\nabla_{\boldsymbol{\theta}}q_k(\boldsymbol{\theta})$ |
| CE(hard) | $-\boldsymbol{e}_{\widehat{y}}^{\top}\log\boldsymbol{q}$ | $\boldsymbol{e}_{\widehat{y}}$ | $\boldsymbol{q} - \boldsymbol{e}_{\widehat{y}}$ | $-\frac{1}{q_{\widehat{y}}(\boldsymbol{\theta})}\nabla_{\boldsymbol{\theta}}q_{\widehat{y}}(\boldsymbol{\theta})$ |
| LDP | $-\log(\boldsymbol{p}^{\top}\boldsymbol{q})$ | $\boldsymbol{e}_{\widehat{y}}$ | $\boldsymbol{q} - \frac{\boldsymbol{p}\odot\boldsymbol{q}}{\boldsymbol{p}^{\top}\boldsymbol{q}}$ | $-\sum_{k=1}^{K}\frac{p_k}{\boldsymbol{p}^{\top}\boldsymbol{q}(\boldsymbol{\theta})}\nabla_{\boldsymbol{\theta}}q_k(\boldsymbol{\theta})$ |

**Relationship between confidence and gradient norm** The loss minimizers and gradients with respect to both logits $\boldsymbol{z} \in \mathbb{R}^K$ and $\boldsymbol{\theta}$ are provided in Table 1, with derivations in Appendix A. Looking at the form of the gradients with respect to $\boldsymbol{\theta}$ in Table 1, we see that the LDP loss implicitly scales the learning rate in an instance-dependent way similarly as the CE loss variants. The scaling factor is smallest when the prediction and target are equal to the same one-hot vector $\boldsymbol{p} = \boldsymbol{q}(\boldsymbol{\theta}) = \boldsymbol{e}_k$ for any $k \in \{1, \ldots, K\}$, and grows larger as $\boldsymbol{p}$ and $\boldsymbol{q}$ become more orthogonal (have non-overlapping support). For the special case $\boldsymbol{p} = \boldsymbol{e}_k$ for some $k \in \{1, \ldots, K\}$, all three loss functions coincide with loss gradient $\boldsymbol{q} - \boldsymbol{e}_k$. For CE(soft) loss it is relatively well known that we get zero gradient only when $\boldsymbol{q} = \boldsymbol{p}$. The minimizer and maximizer of the LDP gradient norm are given in Theorem 1, which tells us that the LDP loss implicitly focuses more on examples with high-confidence and ignores examples with low-confidence. All proofs are in the Appendix.

**Theorem 1.** *Let $\boldsymbol{z} \in \mathbb{R}^K$ be a fixed vector of logits, $\boldsymbol{q}(\boldsymbol{z}) = \mathrm{softmax}(\boldsymbol{z}) \in \Delta$ a prediction for the target $\boldsymbol{p} \in \Delta$, $q_k(\boldsymbol{z})$ the $k$-th entry of vector $\boldsymbol{q}(\boldsymbol{z})$, and $m \in \mathrm{argmin}_k q_k(\boldsymbol{z})$. Then, the Euclidean norm of the gradient of the LDP loss with respect to logits satisfies*

$$0 \leq \|\nabla_{\boldsymbol{z}}[-\log(\boldsymbol{p}^{\top}\boldsymbol{q}(\boldsymbol{z}))]\|_2 \leq \|\boldsymbol{q}(\boldsymbol{z}) - \boldsymbol{e}_m\|_2 = \|\nabla_{\boldsymbol{z}}[-\boldsymbol{e}_m^{\top}\log(\boldsymbol{q}(\boldsymbol{z}))]\|_2. \tag{3}$$

*In addition:*

(a) *The minimum is achieved when the target is the uniform distribution $\boldsymbol{p} = \boldsymbol{1}/K$.*

(b) *The maximum is achieved when the target $\boldsymbol{p}$ is a one-hot vector for label $m$.*

In Theorem 2 we show that the gradient of the LDP loss has a scaling factor that naturally decreases to 0 as the uncertainty of the target distribution increases.

**Theorem 2.** *Let $\boldsymbol{q}(\boldsymbol{\theta}) \in \Delta$ denote the prediction for target $\boldsymbol{p} \in \Delta$ when viewed as a function of the learnable model weights $\boldsymbol{\theta}$. The Euclidean norm of the gradient of the LDP loss with respect to model weights satisfies*

$$\|\nabla_{\boldsymbol{\theta}}[-\log(\boldsymbol{p}^{\top}\boldsymbol{q}(\boldsymbol{\theta}))]\|_2 \leq \frac{\max(\boldsymbol{p}) - \min(\boldsymbol{p})}{\min(\boldsymbol{p})}\sum_{k=1}^{K}||\nabla_{\boldsymbol{\theta}}q_k(\boldsymbol{\theta})||_2. \tag{4}$$

The scaling factor $\frac{\max(\boldsymbol{p})-\min(\boldsymbol{p})}{\min(\boldsymbol{p})}$ depends on the target distribution $\boldsymbol{p}$ only, and tends to 0 when $\boldsymbol{p}$ tends to the uniform distribution. From the theorem's proof, we can tighten the bound by replacing the denominator $\min(\boldsymbol{p})$ in the scaling factor by $\boldsymbol{p}^{\top}\boldsymbol{q}(\boldsymbol{\theta})$. We use the form in the theorem as it gives a simpler interpretation of the effect of target distribution uncertainty on the gradient scale.

Proposition 1 is an analogue of Theorem 1 for several unlabeled data loss functions used in many popular SSL algorithms such as (Sohn et al., 2020; Chen et al., 2023; Wang et al., 2023).

**Proposition 1.** *Let $\boldsymbol{z}, \boldsymbol{q}(\boldsymbol{z})$ and $\boldsymbol{p}$ be defined as in Theorem 1, $p_k$ be the $k$-th element of vector $\boldsymbol{p}$, $M \in \mathrm{argmax}_k p_k$ be any maximizer of $\boldsymbol{p}$, and $w : [1/K, 1] \rightarrow [0, 1]$ be non-decreasing. Then, the Euclidean norm of the gradient with respect to logits of loss functions of the form $\ell(\boldsymbol{q}(\boldsymbol{z}), \boldsymbol{p}) = w(p_M)(-\log q_M(\boldsymbol{z}))$ satisfy the following bound:*

$$w(1/K)\|\boldsymbol{q}(\boldsymbol{z}) - \boldsymbol{e}_M\|_2 \leq \|\nabla_{\boldsymbol{z}}w(p_M)(-\log q_M(\boldsymbol{z}))\|_2 \leq w(1)\|\boldsymbol{q}(\boldsymbol{z}) - \boldsymbol{e}_M\|_2. \tag{5}$$

*In addition:*

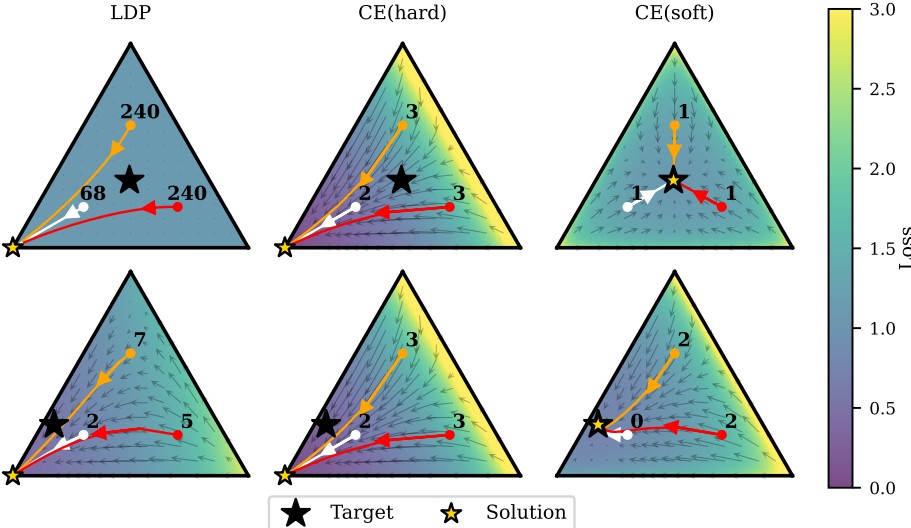

Figure 2: Gradient descent trajectories for LDP, CE(hard), and CE(soft) losses. Explicitly, each trajectory tracks the softmax of the updates $z \leftarrow z - \nabla_z \ell(\mathrm{softmax}(z), p)$, with $p$ being the target, $q^* = \arg\min_q \ell(q, p)$ the solution, and trajectory starting points $z \in \{\log(0.4e_i + 0.2)\}_{i=1}^3$. Number of iterations until convergence $||\mathrm{softmax}(z) - q^*||_2^2 < 0.1$ also indicated. Target 1 $(0.34, 0.33, 0.33)^\top$, target 2 $(0.7, 0.25, 0.05)^\top$ in row 1, 2 respectively.

    *(a) The minimum is achieved when the target is the uniform distribution $p = 1/K$.*

    *(b) The maximum is achieved when the target is any one-hot vector $p = e_k$, $k \in \{1, \ldots, K\}$.*

A special, but common, case of Proposition 1 is when the weighting function is a step function $w(\alpha) = \mathbb{1}(\alpha \geq \tau)$ for some $\tau \in [1/K, 1]$. In this case, like the LDP loss, the lower bound in Equation (5) is zero, but unlike LDP loss, the upper bound is $\|q(z) - e_M\|$, which is bounded from above by $\|q(z) - e_m\|$. Consequently, compared to commonly used loss functions of the form considered in Proposition 1, the LDP loss is able to put extra priority on examples in the extreme case where targets have maximum confidence associated with the class that has the least confidence according to the prediction.

To emphasize the relationship between confidence and gradient norm for different loss functions, we provide visualizations of gradient flow in Figure 2. LDP overall gives a flatter loss landscape compared to both variants of the CE loss, particularly when the target is close to uniform. As a result, gradient updates are generally more conservative for LDP loss, depending on the confidence of the target. A specific example from Figure 2: consider the trajectory beginning at $(0.2, 0.6, 0.2)^\top$ (trajectory starting at top center) with target $(0.34, 0.33, 0.33)^\top$ (top row), it takes 240 gradient descent steps to converge to the solution using LDP loss, but only 3 and 1 iterations for CE(hard) and CE(soft), respectively, despite the high uncertainty in the target. As the entropy of the target decreases, each loss function eventually becomes equivalent. However, the LDP loss tends to produce slightly different trajectories depending on the full target distribution, particularly when compared to CE(hard), which shares the same solution as LDP but ignores the complete target distribution.

We demonstrate how the gradients of the LDP loss are scaled relative to the two variants of the CE loss, CE(Soft) and CE(Hard), in Figure 3. Gradients of the LDP loss are appropriately scaled in the sense that the gradient norm $\|\nabla_z \log(p^\top q(z))\|_2$ is vanishing when the target distribution is close to uniform, close to the gradient norm of CE(Hard) when the target distribution is far from uniform, and is interpolated approximately monotonically in between. The vanishing behavior is expected from Theorem 2 (when applied to the logits), and also note that when $p = e_{\arg\min(q)}$, the CE(Hard) loss gradient $\|\nabla_z e_{\arg\max(p)}^\top \log(q(z))\|_2$ upper-bounds the LDP gradient norm $\|\nabla_z \log(p^\top q(z))\|_2$, by Theorem 1 and illustrated in Figure 3. As a result, when using the LDP loss for consistency regularization in SSL, we do not require any additional regularization constant to ensure the gradient

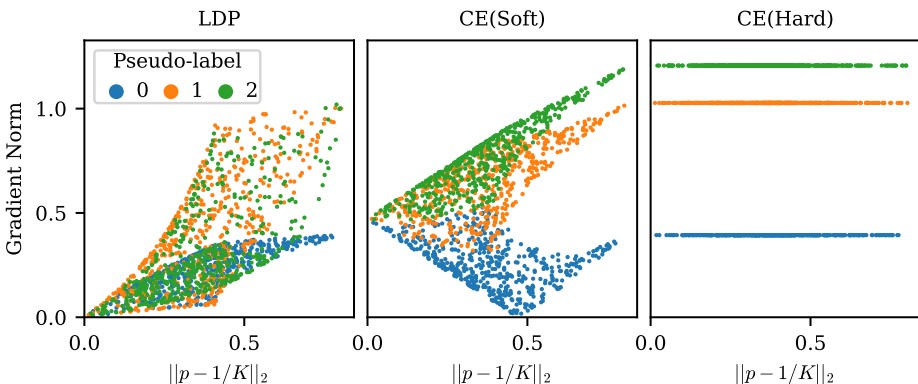

Figure 3: Comparing the distance $\|\boldsymbol{p} - \mathbf{1}/K\|_2$ between target vectors $\boldsymbol{p}$ and the uniform distribution with the norm $\|\nabla_{\boldsymbol{z}} \ell(\boldsymbol{q}(\boldsymbol{z}), \boldsymbol{p})\|_2$ of the gradient with respect to logits for LPD, CE(Soft) and CE(Hard). We fix the prediction vector as $\boldsymbol{q} = (0.7, 0.25, 0.05)^{\top}$. Target vectors are randomly sampled from the probability simplex $\Delta$. Pseudo-labels of each target (index with largest element in $\boldsymbol{p}$) are indicated with different colors. Gradient norms of LDP loss are naturally scaled to be approximately monotonic in the distance between the target distribution and the uniform distribution, with the gradient being exactly equal $\mathbf{0}$ when the target is the uniform distribution. Neither of the CE loss variants have this same monotonic scaling property.

of the consistency regularizer is scaled appropriately relative to the supervised loss. While the gradient norm is generally smallest when both the target and prediction agree on the most likely label for all three loss functions, the CE(Soft) and CE(Hard) loss functions do not share the desirable scaling property where gradient norms are smaller for highly uncertain targets; in particular, even when the target is completely random and thus should be ignored, CE(Soft) and CE(Hard) still have non-zero gradients. This is why when using CE(Soft) or CE(Hard), additional weights are required to manually shrink the gradient norms when the target is close to uniform.

### 4.3 DOTMATCH OBJECTIVE FUNCTION

The DotMatch objective function for a single batch of semi-supervised data is

$$L(\boldsymbol{\theta}) = -\frac{1}{|\mathcal{L}|} \sum_{(\boldsymbol{w}, \boldsymbol{y}) \in \mathcal{L}} \log(\boldsymbol{y}^{\top} \boldsymbol{p}(\boldsymbol{w}; \boldsymbol{\theta})) - \frac{1}{|\mathcal{U}|} \sum_{(\boldsymbol{w}, \boldsymbol{s}) \in \mathcal{U}} \log(\text{DA}(\boldsymbol{p}_{\boldsymbol{w}})^{\top} \boldsymbol{p}(\boldsymbol{s}; \boldsymbol{\theta})), \quad (6)$$

where $\text{DA}(\boldsymbol{p}) = \text{sumnorm}\left((\boldsymbol{p} + \varepsilon\mathbf{1}) \odot \frac{\boldsymbol{\pi} + \varepsilon\mathbf{1}}{\widehat{\boldsymbol{\pi}}_t + \varepsilon\mathbf{1}}\right)$, $\text{sumnorm}(\boldsymbol{x}) = \boldsymbol{x}/\sum_j x_j$, division is element-wise, $\boldsymbol{\pi} \in \Delta$ is the class prior estimated using all available labels, $\widehat{\boldsymbol{\pi}}_t = m\widehat{\boldsymbol{\pi}}_{t-1} + (1 - m)\frac{1}{|\mathcal{U}|}\sum_{(\boldsymbol{w},\boldsymbol{s})\in\mathcal{U}} \boldsymbol{p}_{\boldsymbol{w}}$ is the EMA of the prior estimated using batches of unlabeled data during training, $m \in [0, 1]$ is the EMA momentum, and $\varepsilon$ is a small constant to avoid numerical instabilities which we set to 1e-6 in our experiments. DotMatch inherits the nice properties of the LDP loss, namely consistency, entropy minimization and small gradient norms for low-confidence unlabeled examples. DotMatch also addresses the quantity-quality tradeoff (Chen et al., 2023) by always utilizing all unlabeled examples (high quantity), while implicitly down-weighting the contribution from low-confidence examples (high quality). We compare the performance of DotMatch against state-of-the-art baselines in the next section, and perform an ablation study.

## 5 EXPERIMENTS

**Baselines** We compare DotMatch against two supervised learning baselines: Supervised, which only uses available labeled examples for training to provide a lower bound on performance, and FullySupervised, which uses all labels including true labels for unlabeled examples to give an upper bound on performance. Strong semi-supervised baselines include single model SSL algorithms that have implementations provided with the original papers: FixMatch (Sohn et al., 2020), SoftMatch (Chen et al., 2023), FreeMatch (Wang et al., 2023), FlatMatch (Huang et al., 2023), and InterLUDE (Huang et al., 2024).

Table 2: Test error rates (mean±std% over 3 seeds) of various SSL methods. Best in **bold**, overlapping 95% CI underlined. FullySupervised results for STL10 omitted since not all labels available.

| Dataset | MNIST | | | EMNIST | | | CIFAR10 | | | CIFAR100 | | SVHN | | STL10 | | #1 |
|---|---|---|---|---|---|---|---|---|---|---|---|---|---|---|---|---|
| # Label | 10 | 40 | 250 | 47 | 188 | 1175 | 10 | 40 | 250 | 400 | 2500 | 40 | 250 | 40 | 1000 | |
| Supervised | 67.82±3.26 | 28.96±2.04 | 7.37±0.84 | 85.69±1.00 | 51.94±0.96 | 23.01±0.77 | 82.49±1.24 | 77.01±1.64 | 56.62±0.41 | 89.45±0.86 | 60.05±0.17 | 83.46±2.29 | 25.37±2.79 | 75.49±0.75 | 32.57±1.79 | 0 |
| FixMatch | 44.27±12.83 | 4.78±3.75 | 1.29±0.04 | 45.58±1.11 | 23.88±1.48 | 16.07±0.74 | 78.75±3.96 | 21.03±7.98 | 5.41±0.14 | 51.49±1.40 | 29.33±0.09 | **5.85±4.07** | 2.31±0.04 | 43.65±5.96 | 6.81±0.26 | 1 |
| SoftMatch | 7.31±3.62 | 3.65±2.36 | 1.33±0.06 | 46.42±4.12 | 20.23±0.08 | 15.49±0.41 | 37.75±10.06 | 6.75±0.20 | 5.34±0.09 | 43.12±1.24 | 27.60±0.34 | 21.58±14.34 | 2.54±0.04 | 26.01±6.40 | **6.53±0.06** | 1 |
| FreeMatch | **4.70±2.39** | 2.52±0.85 | 1.34±0.06 | 51.88±6.06 | 20.36±1.89 | 15.83±0.64 | 30.44±7.19 | **5.41±0.04** | 5.28±0.24 | 43.46±2.33 | **27.12±0.29** | 17.94±11.16 | 5.01±1.19 | 28.65±5.97 | 6.54±0.04 | 4 |
| FlatMatch | 24.06±18.17 | **2.30±0.90** | 1.66±0.07 | 52.70±7.27 | 21.03±1.73 | 15.83±0.29 | 29.26±9.44 | 6.35±0.46 | 6.06±0.37 | 46.09±1.94 | 29.13±0.26 | 17.32±4.65 | 9.63±0.76 | **21.00±1.24** | 7.76±0.09 | 2 |
| Interlude | 67.49±4.62 | 13.16±10.22 | **1.10±0.10** | 71.45±7.21 | 32.44±3.10 | 16.98±0.31 | 55.78±8.95 | 27.70±3.63 | 5.75±0.37 | 58.99±2.91 | 46.28±1.76 | 30.59±12.65 | **2.25±0.02** | 42.78±5.35 | 7.87±0.20 | 2 |
| DotMatch (Ours) | 5.74±4.56 | 2.96±2.47 | 1.23±0.17 | **36.20±3.30** | **19.66±1.36** | **15.46±0.59** | 25.32±16.42 | 10.43±2.85 | **5.67±0.28** | **42.96±0.56** | 28.19±0.52 | **15.15±15.45** | 2.49±0.07 | 28.65±4.36 | 7.07±0.08 | 5 |
| FullySupervised | | 0.90±0.04 | | | 11.42±0.11 | | | 4.92±0.09 | | 26.93±0.00 | | 2.46±0.19 | | - | | - |

**Datasets** We choose datasets with a wide variety of input size, number of classes and number of training examples: MNIST (Lecun et al., 1998), EMNIST(balanced) (Cohen et al., 2017), CIFAR10, CIFAR100 (Krizhevsky, 2009), SVHN (Netzer et al., 2011), and STL10 (Coates et al., 2011). A summary of benchmark datasets is provided in Appendix E.1.

**Hyperparameters** All experiments were run using the USB (Wang et al., 2022) classic CV benchmark (equivalent to TorchSSL (Zhang et al., 2021)), apart from FlatMatch and InterLUDE for which we adapted their provided code bases. The architectures used for datasets with color images: WRN-28-2 (Zagoruyko & Komodakis, 2016) for CIFAR10 and SVHN, WRN-28-8 for CIAFR100, WRN-37-2 for STL10 (Zhou et al., 2020). On black and white image datasets we used WRN-10-1 for MNIST and WRN-22-1 for EMNIST, modified to have only one input channel. At test time we make predictions using EMA of the model $\arg\max(p(x; \theta_{EMA}))$. EMA momentum $m$ is set to 0.999. Number of training iterations $T$ for CIFAR10, CIFAR100, SVHN, STL10 is $2 \times 10^5$, similar to USB, except here we train all models from scratch instead of starting with pre-trained models. Number of training iterations for MNIST and EMNIST is $5 \times 10^4$. Learning rate follows the cosine schedule $0.03 \cos(\frac{7\pi t}{16T})$, where $t$ is the training iteration number. Optimizer is SGD with nesterov momentum 0.9. Labeled data batch size 64, unlabeled data batch size 448. See Appendix E.2 for a detailed breakdown of all hyperparameter settings, including algorithm specific hyperparameters used for the baseline methods.

**Benchmark** We compare all algorithms on the datasets with different numbers of labeled examples per class. For example, CIFAR10 with 10 labels means there is 1 labeled example per class. Each algorithm is trained using 3 different random seeds. Following standard practice (Zhang et al., 2021; Chen et al., 2023; Wang et al., 2023; Huang et al., 2023), we report average of best test error rate across all checkpoints in Table 2. Checkpoints occur at every 1000 training iterations. Test error measured after training is completed can be found in Table 7 in Appendix F.

**Results** As shown in Table 2, DotMatch achieves the best average test error rate more often than any of the baseline algorithms (5 for DotMatch versus 4 for second best FreeMatch). It performs particularly well with extremely limited labeled data, either 1 or 4 labels per class. In these settings it achieved the best or equivalent to the best test accuracy 5 times which ties with SoftMatch and FreeMatch. Surprisingly, FixMatch demonstrated the strongest performance on SVHN with 40 labels, with both small average test error and small variance across the random seeds. The small variance is likely due to the simplicity of using a constant confidence threshold rather than a dynamic one. DotMatch, SoftMatch and FreeMatch all had equivalently strong performance on this problem, however with wider confidence intervals.

Notably, DotMatch performs the best across all label settings for the EMNIST dataset, which, to the best of our knowledge, has not been used in such an SSL benchmark previously. In particular, DotMatch is significantly better than all baseline methods on EMNIST with 1 labeled example per class. This is likely because DotMatch has a significant advantage of not requiring tuning for any algorithm specific hyperparameters, whereas we copied the recommended algorithm specific hyperparameters for baseline methods from CIFAR10 since there was not enough labeled data available to do proper tuning.

**Runtime** We compared the runtime per iteration of DotMatch against baseline methods and report the results in Table 8 in the Appendix. All algorithms were implemented in a unified code base and run on identical hardware. DotMatch is consistently the fastest across all datasets, with FixMatch generally being the second fastest, followed by FreeMatch then SoftMatch.

Table 3: Test error rates (mean±std over 3 seeds) for different variations of DotMatch on EMNIST with varying amounts of labeled data. Best in **bold**, overlapping 95% CI underlined.

| Loss | DA | EMNIST | | |
|------|----|----|----|----|
| | | 47 | 188 | 1175 |
| CE(soft) | ✗ | $65.87_{\pm0.48}$ | $28.84_{\pm0.76}$ | $17.73_{\pm0.95}$ |
| CE(hard) | ✗ | $50.77_{\pm2.80}$ | $27.20_{\pm2.32}$ | $16.95_{\pm0.58}$ |
| LDP | ✗ | $56.73_{\pm1.77}$ | $24.99_{\pm1.92}$ | $\underline{15.61_{\pm0.60}}$ |
| CE(soft) | ✓ | $79.21_{\pm1.23}$ | $43.28_{\pm2.45}$ | $21.09_{\pm0.72}$ |
| CE(hard) | ✓ | $46.21_{\pm2.44}$ | $26.03_{\pm2.06}$ | $16.90_{\pm0.34}$ |
| LDP | ✓ | $\mathbf{36.20_{\pm3.30}}$ | $\mathbf{19.66_{\pm1.36}}$ | $\mathbf{15.46_{\pm0.59}}$ |

Overall, DotMatch generally performs at least as well as the state-of-art baselines on most problems, while being simpler, faster, and not requiring any tuning of algorithm specific hyperparameters like regularization strength or confidence threshold schedule.

### 5.1 ABLATION

We conduct experiments on the utility of the components of DotMatch, namely the choice of loss function and use of DA. We compare three different choices for the unlabeled loss function $\ell_u$: CE(soft), CE(hard), and LDP, each with and without DA applied to the targets. We train on EMNIST using the same experiment settings from the main benchmark.

As shown in Table 3, DotMatch (LDP+DA) achieves the best result in every case, with only LDP without DA matching the performance when there are 1175 labeled examples, or 25 per class. Other than LDP loss, CE(hard) with DA generally gives the second best results across all settings, which validates that entropy minimization (in this case via one-hot pseudo-labeling) is an important design consideration for SSL algorithms, which is lacking when using CE(soft). Adding DA consistently gives performance at least as good, if not better, when combined with CE(hard) or LDP loss, but hurts performance when using CE(soft). We suspect this is because it is most sensitive to changes in the target since the minimizer also changes, whereas for CE(hard) and LDP the minimizer only changes if the pseudo-label changes, which is not always the case when using DA.

## 6 CONCLUSION

We proposed the log dot product loss function for consistency regularization in SSL, and used it in the novel algorithm DotMatch. Theoretical analysis shows that DotMatch exhibits desirable qualities of an SSL algorithm, namely consistency, entropy minimization and small gradient norms for low-confidence unlabeled examples. Notably, no algorithm specific hyperparameters need to be tuned for DotMatch, making it much simpler than existing algorithms. Minimizing the number of hyperparameters in SSL algorithms is particularly important since there is often insufficient labeled examples available for validation purposes. Our DotMatch demonstrated strong performance in an extensive benchmark against state-of-the-art baselines, and we showed that the proposed configuration is the best among natural alternatives in the ablation study.

Since the unlabeled data loss used in DotMatch based on LDP loss does not require any labels, and the properties of consistency, entropy minimization and DA are desirable for any classifier, it is also suitable as a plug-in regularizer in other weakly supervised learning (WSL) (Zhou, 2017) settings to help improve performance without introducing additional hyperparameters. For example, in learning from complementary labels (Ishida et al., 2017; Feng et al., 2020), noisy labels (Frenay & Verleysen, 2014; Song et al., 2023), partial labels (Grandvalet & Bengio, 2004b; Lv et al., 2024), positive-unlabeled data (Elkan & Noto, 2008), similarity labels (Bao et al., 2018; Hsu et al., 2019) or unlabeled data with label proportions (Tang et al., 2023). A limitation of this study is that we did not consider SSL problems with large class imbalances, mainly because it is difficult to estimate the true imbalanced label distribution in interesting and challenging settings with extremely scarce labeled data. In principle, DotMatch automatically adapts to imbalanced scenarios, provided the labeled data label distribution is an accurate reflection of the true label distribution for unlabeled data. We leave the thorough investigation into imbalanced SSL problems and use of LDP in other WSL settings to future work.

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

## A DERIVATION OF RESULTS FROM TABLE 1

Let $\boldsymbol{q}(\boldsymbol{z}) = \mathrm{softmax}(\boldsymbol{z}) \in \Delta$ be the predicted label distribution for the target $\boldsymbol{p} \in \Delta$, $\boldsymbol{z} \in \mathbb{R}^K$ are logits, and $\widehat{y} = \mathrm{argmax}(\boldsymbol{p})$. We first derive the gradient of a single element $\nabla_{\boldsymbol{z}} q_i(\boldsymbol{z})$, which we will use to simplify other calculations. To do this, consider the partial derivative of a single element $q_i$ with respect to a single element of $\boldsymbol{z}$, $z_j$.

$$
\begin{aligned}
\frac{\partial q_i}{\partial z_j} &= q_i \frac{\partial \log q_i}{\partial z_j} \\
&= q_i \frac{\partial}{\partial z_j} \log \left( \frac{\exp(z_i)}{\sum_{k=1}^K \exp(z_k)} \right) \\
&= q_i \frac{\partial}{\partial z_j} \left( z_i - \log \left( \sum_{k=1}^K \exp(z_k) \right) \right) \\
&= q_i \left( \mathbb{1}(i = j) - \frac{\exp(z_j)}{\sum_{k=1}^K \exp(z_k)} \right) \\
&= q_i \left( \mathbb{1}(i = j) - q_j \right).
\end{aligned}
$$

The gradient of $q_i$ with respect to $\boldsymbol{z}$ is thus

$$
\nabla_{\boldsymbol{z}} q_i(\boldsymbol{z}) = q_i(\boldsymbol{z})(\boldsymbol{e}_i - \boldsymbol{q}(\boldsymbol{z})),
$$

and the full Jacobian matrix

$$
\nabla_{\boldsymbol{z}} \boldsymbol{q}(\boldsymbol{z}) = \mathrm{diag}(\boldsymbol{q}(\boldsymbol{z})) - \boldsymbol{q}(\boldsymbol{z})\boldsymbol{q}(\boldsymbol{z})^\top,
$$

where $\mathrm{diag}(\boldsymbol{q})$ is the diagonal matrix with elements of $\boldsymbol{q}$ on the diagonal.

**CE(soft)** The loss is

$$
\begin{aligned}
\ell(\boldsymbol{q}, \boldsymbol{p}) &= -\boldsymbol{p}^\top \log \boldsymbol{q} \\
&= \boldsymbol{p}^\top \log \left( \frac{\boldsymbol{p}}{\boldsymbol{q}} \right) - \boldsymbol{p}^\top \log \boldsymbol{p} \\
&= \mathrm{KL}(\boldsymbol{p} \| \boldsymbol{q}) + \mathrm{Ent}(\boldsymbol{p}).
\end{aligned}
$$

Minimizing the CE loss over $\boldsymbol{q}$ is equivalent to minimizing the KL-divergence, which happens when $\boldsymbol{q} = \boldsymbol{p}$. Now we derive the gradient of the CE(soft) loss with respect to logits:

$$\nabla_{\boldsymbol{z}} - \boldsymbol{p}^\top \log \boldsymbol{q}(\boldsymbol{z}) = -\nabla_{\boldsymbol{z}} \sum_{k=1}^{K} p_k \log q_k(\boldsymbol{z})$$

$$= -\sum_{k=1}^{K} p_k \nabla_{\boldsymbol{z}} \log q_k(\boldsymbol{z})$$

$$= -\sum_{k=1}^{K} p_k \frac{\nabla_{\boldsymbol{z}} q_k(\boldsymbol{z})}{q_k(\boldsymbol{z})}$$

$$= -\sum_{k=1}^{K} p_k \frac{q_k(\boldsymbol{z})(\boldsymbol{e}_k - \boldsymbol{q}(\boldsymbol{z}))}{q_k(\boldsymbol{z})}$$

$$= -\sum_{k=1}^{K} p_k(\boldsymbol{e}_k - \boldsymbol{q}(\boldsymbol{z}))$$

$$= \boldsymbol{q}(\boldsymbol{z}) - \boldsymbol{p}.$$

Now consider $\boldsymbol{q}$ as a general function of parameters $\boldsymbol{\theta}$. The gradient with respect to $\boldsymbol{\theta}$ is:

$$\nabla_{\boldsymbol{\theta}} - \boldsymbol{p}^\top \log \boldsymbol{q}(\boldsymbol{\theta}) = -\nabla_{\boldsymbol{\theta}} \sum_{k=1}^{K} p_k \log q_k(\boldsymbol{\theta})$$

$$= -\sum_{k=1}^{K} \frac{p_k}{q_k(\boldsymbol{\theta})} \nabla_{\boldsymbol{\theta}} q_k(\boldsymbol{\theta}).$$

CE(HARD)

The loss is $\ell(\boldsymbol{q}, \boldsymbol{p}) = -\boldsymbol{e}_{\widehat{y}}^\top \log \boldsymbol{q} = -\log q_{\widehat{y}}$. This is minimized when $q_{\widehat{y}}$ is maximized, that is, $\boldsymbol{q}$ assigns all probability mass to class $\widehat{y}$, $\boldsymbol{q} = \boldsymbol{e}_{\widehat{y}}$.

Since CE(hard) is a special case of CE(soft) with target $\boldsymbol{p} = \boldsymbol{e}_{\widehat{y}}$, we can plug this value into the gradients for CE(soft) to get the gradients of CE(hard):

$$\nabla_{\boldsymbol{z}} - \boldsymbol{e}_{\widehat{y}}^\top \log \boldsymbol{q}(\boldsymbol{z}) = \boldsymbol{q}(\boldsymbol{z}) - \boldsymbol{e}_{\widehat{y}},$$

$$\nabla_{\boldsymbol{\theta}} - \boldsymbol{e}_{\widehat{y}}^\top \log \boldsymbol{q}(\boldsymbol{\theta}) = -\frac{1}{q_{\widehat{y}}(\boldsymbol{\theta})} \nabla_{\boldsymbol{\theta}} q_{\widehat{y}}(\boldsymbol{\theta}).$$

LDP

The LDP loss is $\ell(\boldsymbol{q}, \boldsymbol{p}) = -\log(\boldsymbol{p}^\top \boldsymbol{q}) = -\log(\sum_{k=1}^{K} p_k q_k)$. To minimize this loss over $\boldsymbol{q}$, we need to maximize the dot product $\sum_{k=1}^{K} p_k q_k$. This is achieved when $\boldsymbol{q}$ puts all of its mass on the largest elements of $\boldsymbol{p}$,

$$\underset{\boldsymbol{q} \in \Delta}{\operatorname{argmin}} - \log(\boldsymbol{p}^\top \boldsymbol{q}) = \{\boldsymbol{q} \in \Delta \; : \; q_k = 0 \, \forall \, k \notin \operatorname{argmax}(\boldsymbol{p})\}.$$

In the usual case where the largest element in $\boldsymbol{p}$ is unique, we have $\underset{\boldsymbol{q} \in \Delta}{\operatorname{argmin}} - \log(\boldsymbol{p}^\top \boldsymbol{q}) = \boldsymbol{e}_{\widehat{y}}$.

Now compute the gradient of LDP loss with respect to logits:

$$\nabla_{\boldsymbol{z}} - \log(\boldsymbol{p}^\top \boldsymbol{q}(\boldsymbol{z})) = -\frac{\nabla_{\boldsymbol{z}} \boldsymbol{q}(\boldsymbol{z})^\top \boldsymbol{p}}{\boldsymbol{p}^\top \boldsymbol{q}(\boldsymbol{z})}$$

$$= -\frac{(\operatorname{diag}(\boldsymbol{q}(\boldsymbol{z})) - \boldsymbol{q}(\boldsymbol{z})\boldsymbol{q}(\boldsymbol{z})^\top)\boldsymbol{p}}{\boldsymbol{p}^\top \boldsymbol{q}(\boldsymbol{z})}$$

$$= -\frac{\boldsymbol{p} \odot \boldsymbol{q}(\boldsymbol{z}) - \boldsymbol{q}(\boldsymbol{z})\boldsymbol{q}(\boldsymbol{z})^\top \boldsymbol{p}}{\boldsymbol{p}^\top \boldsymbol{q}(\boldsymbol{z})}$$

$$= \boldsymbol{q}(\boldsymbol{z}) - \frac{\boldsymbol{p} \odot \boldsymbol{q}(\boldsymbol{z})}{\boldsymbol{p}^\top \boldsymbol{q}(\boldsymbol{z})}.$$

The gradient with respect to $\boldsymbol{\theta}$ is

$$
\begin{aligned}
\nabla_{\boldsymbol{\theta}} - \log(\boldsymbol{p}^\top \boldsymbol{q}(\boldsymbol{\theta})) &= -\nabla_{\boldsymbol{\theta}} \log \left( \sum_{k=1}^K p_k q_k(\boldsymbol{\theta}) \right) \\
&= -\frac{\nabla_{\boldsymbol{\theta}} \sum_{k=1}^K p_k q_k(\boldsymbol{\theta})}{\sum_{k=1}^K p_k^\top q_k(\boldsymbol{\theta})} \\
&= -\frac{\sum_{k=1}^K p_k \nabla_{\boldsymbol{\theta}} q_k(\boldsymbol{\theta})}{\sum_{k=1}^K p_k q_k(\boldsymbol{\theta})} \\
&= -\sum_{k=1}^K \frac{p_k}{\boldsymbol{p}^\top \boldsymbol{q}(\boldsymbol{\theta})} \nabla_{\boldsymbol{\theta}} q_k(\boldsymbol{\theta}).
\end{aligned}
$$

## B  PROOF OF THEOREM 1

**Theorem 1.** *Let $\boldsymbol{z} \in \mathbb{R}^K$ be a fixed vector of logits, $\boldsymbol{q}(\boldsymbol{z}) = \mathrm{softmax}(\boldsymbol{z}) \in \Delta$ a prediction for the target $\boldsymbol{p} \in \Delta$, $q_k(\boldsymbol{z})$ the $k$-th entry of vector $\boldsymbol{q}(\boldsymbol{z})$, and $m \in \arg\min_k q_k(\boldsymbol{z})$. Then, the Euclidean norm of the gradient of the LDP loss with respect to logits satisfies*

$$
0 \le \|\nabla_{\boldsymbol{z}}[-\log(\boldsymbol{p}^\top \boldsymbol{q}(\boldsymbol{z}))]\|_2 \le \|\boldsymbol{q}(\boldsymbol{z}) - \boldsymbol{e}_m\|_2 = \|\nabla_{\boldsymbol{z}}[-\boldsymbol{e}_m^\top \log(\boldsymbol{q}(\boldsymbol{z}))]\|_2. \tag{3}
$$

*In addition:*

  (a) *The minimum is achieved when the target is the uniform distribution $\boldsymbol{p} = \mathbf{1}/K$.*

  (b) *The maximum is achieved when the target $\boldsymbol{p}$ is a one-hot vector for label $m$.*

*Proof.* We use the short-hand notation $\boldsymbol{q}$ to represent $\boldsymbol{q}(\boldsymbol{z})$. If there exist a probability vector $\boldsymbol{p}$ that is a stationary point of the LDP loss, then this will minimize the gradient norm. To find the stationary point, set the gradient equal to the zero vector and rearrange:

$$
\mathbf{0} = \boldsymbol{q} - \frac{\boldsymbol{p} \odot \boldsymbol{q}}{\boldsymbol{p}^\top \boldsymbol{q}},
$$

$$
\boldsymbol{p}^\top \boldsymbol{q} \boldsymbol{q} = \boldsymbol{p} \odot \boldsymbol{q}.
$$

We have equality between two vectors, so each element should be equal, that is, $\boldsymbol{p}^\top \boldsymbol{q} q_k = p_k q_k$ for all $k$. The equation is satisfied when $q_k = 0$ and corresponding $p_k$ can be arbitrary. However, we assume $\boldsymbol{q}$ is a softmax vector, so no element can be exactly equal to zero. In the case $q_k \ne 0$, we can divide both sides by $q_k$, giving $\boldsymbol{p}^\top \boldsymbol{q} = p_k$. This implies $p_k$ is constant for all $k$, or $\boldsymbol{p} = \mathbf{1}/K$ is the uniform distribution.

To find the $\boldsymbol{p}$ that maximizes the norm of the gradient $\|\boldsymbol{q} - \boldsymbol{p} \odot \boldsymbol{q}/\boldsymbol{p}^\top \boldsymbol{q}\|_2$, note that $\boldsymbol{x} = \boldsymbol{p} \odot \boldsymbol{q}/\boldsymbol{p}^\top \boldsymbol{q}$ is a probability vector. This implies that (a) $\|\boldsymbol{x}\|_2^2 \le 1$, where equality holds when $\boldsymbol{x}$ is a one-hot vector; and (b) $\boldsymbol{q}^\top \boldsymbol{x} \ge q_m$, where equality holds when $\boldsymbol{x}$ is the one-hot vector $\boldsymbol{e}_m$. Hence we have

$$
\|\boldsymbol{q} - \boldsymbol{x}\|_2^2 = \|\boldsymbol{q}\|_2^2 - 2\boldsymbol{q}^\top \boldsymbol{x} + \|\boldsymbol{x}\|_2^2 \le \|\boldsymbol{q}\|_2^2 - 2q_m + 1 = \|\boldsymbol{q} - \boldsymbol{e}_m\|_2^2,
$$

where the maximum is achieved when $\boldsymbol{x} = \boldsymbol{e}_m$; equivalently, when $\boldsymbol{p} \odot \boldsymbol{q}/\boldsymbol{p}^\top \boldsymbol{q} = \boldsymbol{e}_m$, or $\boldsymbol{p} = \boldsymbol{e}_m$.

$\square$

## C  PROOF OF THEOREM 2

**Theorem 2.** *Let $\boldsymbol{q}(\boldsymbol{\theta}) \in \Delta$ denote the prediction for target $\boldsymbol{p} \in \Delta$ when viewed as a function of the learnable model weights $\boldsymbol{\theta}$. The Euclidean norm of the gradient of the LDP loss with respect to model weights satisfies*

$$
\|\nabla_{\boldsymbol{\theta}}[-\log(\boldsymbol{p}^\top \boldsymbol{q}(\boldsymbol{\theta}))]\|_2 \le \frac{\max(\boldsymbol{p}) - \min(\boldsymbol{p})}{\min(\boldsymbol{p})} \sum_{k=1}^K \|\nabla_{\boldsymbol{\theta}} q_k(\boldsymbol{\theta})\|_2. \tag{4}
$$

*Proof.* The gradient of the LDP loss with respect to model weights $\boldsymbol{\theta}$ can be read from Table 1 row 3 column 5:

$$\nabla_{\boldsymbol{\theta}}[-\log(\boldsymbol{p}^\top \boldsymbol{q}(\boldsymbol{\theta}))] = -\sum_{k=1}^{K} \frac{p_k}{\boldsymbol{p}^\top \boldsymbol{q}(\boldsymbol{\theta})} \nabla_{\boldsymbol{\theta}} q_k(\boldsymbol{\theta}).$$

Since $\sum_{k=1}^{K} q_k(\boldsymbol{\theta}) = 1$, we have $\nabla_{\boldsymbol{\theta}} \sum_{k=1}^{K} q_k(\boldsymbol{\theta}) = \mathbf{0}$, and

$$\nabla_{\boldsymbol{\theta}}[-\log(\boldsymbol{p}^\top \boldsymbol{q}(\boldsymbol{\theta}))] = -\frac{1}{\boldsymbol{p}^\top \boldsymbol{q}(\boldsymbol{\theta})} \sum_{k=1}^{K} (p_k - \min(\boldsymbol{p})) \nabla_{\boldsymbol{\theta}} q_k(\boldsymbol{\theta}).$$

Applying the triangle inequality to the norm of the gradient gives

$$\left\| \nabla_{\boldsymbol{\theta}}[-\log(\boldsymbol{p}^\top \boldsymbol{q}(\boldsymbol{\theta}))] \right\|_2 \leq \frac{1}{\boldsymbol{p}^\top \boldsymbol{q}(\boldsymbol{\theta})} \sum_{k=1}^{K} (p_k - \min(\boldsymbol{p})) \left\| \nabla_{\boldsymbol{\theta}} q_k(\boldsymbol{\theta}) \right\|_2. \tag{7}$$

We can simplify the upper bound in Equation (7) by applying $\boldsymbol{p}^\top \boldsymbol{q}(\boldsymbol{\theta}) \geq \min(\boldsymbol{p})$ and $p_k \leq \max(\boldsymbol{p})$ to give the result

$$\|\nabla_{\boldsymbol{\theta}}[-\log(\boldsymbol{p}^\top \boldsymbol{q}(\boldsymbol{\theta}))]\|_2 \leq \frac{\max(\boldsymbol{p}) - \min(\boldsymbol{p})}{\min(\boldsymbol{p})} \sum_{k=1}^{K} \left\| \nabla_{\boldsymbol{\theta}} q_k(\boldsymbol{\theta}) \right\|_2.$$

$\square$

## D    PROOF OF PROPOSITION 1

**Proposition 1.** *Let $\boldsymbol{z}, \boldsymbol{q}(\boldsymbol{z})$ and $\boldsymbol{p}$ be defined as in Theorem 1, $p_k$ be the $k$-th element of vector $\boldsymbol{p}$, $M \in \arg\max_k p_k$ be any maximizer of $\boldsymbol{p}$, and $w : [1/K, 1] \to [0, 1]$ be non-decreasing. Then, the Euclidean norm of the gradient with respect to logits of loss functions of the form $\ell(\boldsymbol{q}(\boldsymbol{z}), \boldsymbol{p}) = w(p_M)(-\log q_M(\boldsymbol{z}))$ satisfy the following bound:*

$$w(1/K) \|\boldsymbol{q}(\boldsymbol{z}) - \boldsymbol{e}_M\|_2 \leq \|\nabla_{\boldsymbol{z}} w(p_M)(-\log q_M(\boldsymbol{z}))\|_2 \leq w(1) \|\boldsymbol{q}(\boldsymbol{z}) - \boldsymbol{e}_M\|_2. \tag{5}$$

*In addition:*

(a) *The minimum is achieved when the target is the uniform distribution $\boldsymbol{p} = \mathbf{1}/K$.*

(b) *The maximum is achieved when the target is any one-hot vector $\boldsymbol{p} = \boldsymbol{e}_k$, $k \in \{1, \dots, K\}$.*

*Proof.* The gradient of the loss can be read from Table 1 and scaled by the weight function, $\nabla_{\boldsymbol{z}} \ell(\boldsymbol{q}(\boldsymbol{z}), \boldsymbol{p}) = w(p_M)(\boldsymbol{q}(\boldsymbol{z}) - \boldsymbol{e}_M)$. When viewed as a function of $\boldsymbol{p}$, the norm of this gradient $w(p_M)\|\boldsymbol{q}(\boldsymbol{z}) - \boldsymbol{e}_M\|_2$ takes on its extreme values when the weight function $w$ is minimized or maximized. Since $w$ is non-decreasing, it attains its minimum value at $1/K$ when $\boldsymbol{p} = \mathbf{1}/K$ is the uniform distribution, and attains is maximum value when $\boldsymbol{p}$ is a one-hot vector with $p_M = 1$. $\square$

## E    EXPERIMENT SETTINGS

### E.1    DATASETS

We provide some summary statistics about each of the datasets included in the benchmark in Table 4.

### E.2    HYPERPARAMETERS

To aid in reproducibility, we present all hyperparameters used in the experiments. Algorithm-independent hyperparameters are provided in Table 5.

For the black and white datasets MNIST and EMNIST, the models are modified to have one input channel rather than three, and only the subset of the augmentations in RandAugment that do not rely on color are used.

For algorithm dependent hyperparameters we generally copy values from the original papers: Fix-Match $\tau = 0.95$, SoftMatch uses adaptive $\mu_t$ and $\sigma_t$ in the weight function, FreeMatch fairness

Table 4: Benchmark Datasets.

| Dataset | $n$ train | $n$ test | $K$ | $d$ |
|---|---|---|---|---|
| MNIST (Lecun et al., 1998) | 60 000 | 10 000 | 10 | $28 \times 28$ |
| EMNIST (Cohen et al., 2017) | 112 800 | 18 800 | 47 | $28 \times 28$ |
| CIFAR10 (Krizhevsky, 2009) | 50 000 | 10 000 | 10 | $3 \times 32 \times 32$ |
| CIFAR100 (Krizhevsky, 2009) | 50 000 | 10 000 | 100 | $3 \times 32 \times 32$ |
| SVHN (Netzer et al., 2011) | 604 388 | 26 032 | 10 | $3 \times 32 \times 32$ |
| STL10 (Coates et al., 2011) | 105 000 | 8 000 | 10 | $3 \times 96 \times 96$ |
| ImageNet (Deng et al., 2009) | 1 281 167 | 50 000 | 1 000 | $3 \times 224 \times 224$ |

Table 5: Algorithm independent hyperparameters.

| Dataset | MNIST | EMNIST | CIFAR-10 | CIFAR-100 | STL-10 | SVHN | ImageNet |
|---|---|---|---|---|---|---|---|
| Model | WRN-10-1 | WRN-22-1 | WRN-28-2 | WRN-28-8 | WRN-37-2 | WRN-28-2 | ResNet50 |
| Train iters | 50 000 | 50 000 | 200 000 | 200 000 | 200 000 | 200 000 | 1 048 576 |
| Weight decay | | 5e-4 | | 1e-3 | | 5e-4 | 3e-4 |
| Labeled batch size | | | | 64 | | | 128 |
| Unlabeled batch size | | | | 64 | | | 128 |
| Learning rate | | | | $0.03 \cos(\frac{7\pi t}{16T})$ | | | |
| SGD momentum | | | | 0.9 | | | |
| EMA momentum | | | | 0.999 | | | |
| Weak Augmentation | | | | Random Crop and Flip | | | |
| Strong Augmentation | | | | RandAugment | | | |

loss weight $\lambda_{SAF} = 0.01$ for MNIST, EMNIST, CIFAR10-10, CIFAR100-400, STL10-40, SVHN, and $\lambda_{SAF} = 0.05$ for all other cases. FlatMatch perturbation magnitude $\rho = 0.1$, InterLUDE delta consistency loss weight $\lambda_{DC} = 1.0$, fusion strength $\alpha = 0.1$. All experiments were run on a cluster containing NVIDIA V100, A100 and A30 gpus.

The number of learnable parameters for each different architecture are provided in Table 6.

Table 6: Architectures.

| Name | #Params |
|---|---|
| WRN-10-1 | 77 578 |
| WRN-22-1 | 274 415 |
| WRN-28-2 | 1 467 626 |
| WRN-28-8 | 23 401 028 |
| WRN-37-2 | 5 929 450 |
| ResNet50 | 25 557 032 |

# F ADDITIONAL EXPERIMENT RESULTS

The test accuracy recorded after training is completed is presented in Table 7. The conclusions are similar as for Table 2. Overall DotMatch achieves the best or equivalent to the best performance 11 times out of 15 experiments, only matched by other state-of-the-art methods SoftMatch and FreeMatch. DotMatch either achieves the best or equivalent to the best performance in all of the most challenging settings with the least amount of labeled data, the only algorithm to do so. It also has consistently strong performance on MNIST and EMNIST, which are datasets not typically included in benchmarks from other works.

We compared the runtime of DotMatch against baseline methods and report the results in Table 8. Time is recorded as the average seconds per iteration for the first 1000 training iterations. All algorithms are implemented using the USB code base, and hardware is kept identical: 1 NVIDIA A100 40GB, 128GB RAM. DotMatch is consistently the fastest algorithm across all datasets, with FixMatch generally being the second fastest, followed by FreeMatch then SoftMatch. Interestingly, FixMatch was the slowest on ImageNet out of the four algorithms, but the rest of the rankings on ImageNet remain the same as for other datasets.

Table 7: Final test error rates (mean±std% over 3 seeds) of various SSL methods recorded after training completed. Best in **bold**, overlapping 95% CI underlined. FullySupervised results for STL10 omitted since not all labels available.

| Dataset | MNIST | | | EMNIST | | | CIFAR10 | | | CIFAR100 | | SVHN | | STL10 | |
|---|---|---|---|---|---|---|---|---|---|---|---|---|---|---|---|
| # Label | 10 | 40 | 250 | 47 | 188 | 1175 | 10 | 40 | 250 | 400 | 2500 | 40 | 250 | 40 | 1000 |
| Supervised | 71.31±5.54 | 30.36±2.97 | 8.98±1.37 | 85.69±1.00 | 52.68±1.60 | 23.30±0.65 | 84.05±1.46 | 79.15±1.15 | 62.22±6.36 | 89.88±1.23 | 61.63±0.52 | 87.37±1.55 | 32.68±8.72 | 77.95±1.93 | 33.81±1.55 |
| FixMatch | 44.41±12.81 | 4.96±3.98 | 1.34±0.04 | 48.24±2.07 | 24.14±1.37 | 16.11±0.76 | 85.84±6.01 | 22.10±8.33 | 5.46±0.16 | 52.58±1.33 | 29.93±0.12 | **5.88±4.10** | 2.37±0.08 | 48.00±8.67 | 7.07±0.38 |
| SoftMatch | 8.51±4.93 | 3.69±2.39 | 1.38±0.04 | 49.32±2.91 | **20.60±0.39** | 15.61±0.34 | 42.17±11.95 | 6.91±0.29 | 5.51±0.07 | **45.65±1.33** | 28.73±0.41 | 24.07±12.61 | 2.55±0.04 | 34.71±9.71 | **6.60±0.11** |
| FreeMatch | **4.97±2.49** | 2.88±1.14 | 1.37±0.06 | 54.81±4.22 | 21.09±2.42 | 15.86±0.59 | 32.56±7.83 | **5.51±0.09** | **5.39±0.21** | 46.57±2.48 | **28.47±0.28** | 28.77±16.97 | 13.80±1.43 | 34.32±9.76 | 6.64±0.07 |
| FlatMatch | 26.49±21.15 | **2.45±1.06** | 1.70±0.06 | 53.84±6.27 | 21.33±1.66 | 15.84±0.27 | 30.92±11.46 | 6.54±0.56 | 6.10±0.34 | 47.46±1.72 | 29.58±0.11 | 21.45±7.33 | 13.69±0.55 | **21.12±1.32** | 7.94±0.18 |
| Interlude | 67.51±4.62 | 13.30±10.37 | **1.10±0.10** | 73.32±8.43 | 32.89±2.96 | 17.08±0.27 | 55.87±8.98 | 28.19±4.07 | 5.80±0.38 | 60.41±3.08 | 48.36±1.62 | 33.25±15.24 | **2.26±0.03** | 43.66±5.86 | 7.92±0.13 |
| DotMatch | 6.40±5.46 | 2.98±2.46 | 1.25±0.18 | **44.76±1.82** | 20.83±1.16 | **15.55±0.57** | 29.09±18.63 | 10.71±2.77 | 5.86±0.33 | 46.04±0.65 | 29.51±0.58 | 16.48±14.47 | 2.53±0.08 | 36.40±12.81 | 7.23±0.09 |
| FullySupervised | | 0.93±0.06 | | | 11.42±0.11 | | | 5.05±0.12 | | 26.93±0.00 | | 2.49±0.20 | | - | |

Table 8: Training time (seconds/iteration) for various SSL methods. Best in **bold**, second best in underline. Algorithms are implemented using a unified codebase and run using identical hardware.

| Dataset | MNIST | EMNIST | CIFAR10 | CIFAR100 | SVHN | STL10 | ImageNet |
|---|---|---|---|---|---|---|---|
| # Label | 250 | 188 | 250 | 400 | 250 | 1000 | 100 000 |
| FixMatch | 0.284 | 0.303 | 0.415 | 0.556 | 0.412 | 0.679 | 1.758 |
| SoftMatch | 0.289 | 0.313 | 0.444 | 0.666 | 0.443 | 0.763 | 1.620 |
| FreeMatch | 0.286 | 0.305 | 0.418 | 0.563 | 0.416 | 0.694 | 1.579 |
| DotMatch | **0.280** | **0.293** | **0.393** | **0.460** | **0.396** | **0.624** | **1.538** |

