# OpenReview forum: "DotMatch: Simplified Semi-Supervised Learning with the Log Dot Product Loss"
_ICLR.cc/2026/Conference — Submitted to ICLR 2026_

### Official Review · Reviewer_YaiK · 2025-10-31

**Soundness:** 3
**Presentation:** 3
**Contribution:** 2
**Rating:** 6
**Confidence:** 5

**Summary:**

This paper introduces a new method from Semi-Supervised Learning (SSL) called DotMatch.
The goal of this method is to simplify existing methods by removing user-specified hyper-parameters such a label confidence threshold or such as the weighting for the unlabeled (which in many method is set to 1 anyway).
This is achieved by replacing the cross-entropy (CE) loss $\text{CE}(x,y)=-\sum_i y_i\log(x_i)$ where $y$ is the expected label probability vector and $x$ the actual label probability vector, with a new loss called Log Dot Product (LDP) loss $\text{LDP}(x,y)=-\log(\sum_i x_i y_i)$.


Using this simple modification, the method adapts the classical unlabeled SSL loss for a neural network classifier $p(.;\theta)$ to $$L_{u}(\theta)=\text{LDP}(p(s(x);\theta),\text{nograd}(\text{DA}(p(w(x);\theta))))$$ where $w,s$ are respectively weak and strong augmentations.
Here $\text{DA}$ stands for distribution alignement and is a reweighting of the label probability vector obtained from $w(x)$ by an approximation of the ratio $\mathbb{E}_{y\sim\mathcal{L}}[y] / \hat{p}_y=\mathbb{E}_{x\sim\mathcal{U}}[p(w(x);\theta)]$ as introduced by ReMixMatch.

Results are evaluated with SotA baslines for MNIST, EMNIST, CIFAR10, CIFAR100, SVHN and STL confirming the method is competitive for various sizes of labeled subsets.

**Strengths:**

1. The method is simple and elegant.
2. The method uses all the unlabeled data in the batch rather than zero-ing low-confidence predicted labels unlike previous methods.
3. The paper is well written and easy to follow for the most part.

**Weaknesses:**

1. The definition for $\texttt{sumnorm}$ (line 364) appears to be missing, I assume it must be $\text{sumnorm}(x)=x/\sum_i x_i$.
2. The definition for $\epsilon$ also appears to be missing (line 364), I assume it must be a small constant to avoid division by 0.

**Questions:**

1. What value of $\epsilon$ did you use?
2. Is $\epsilon$ a function of the number of classes?

---

> ### Author Response · Authors · 2025-11-26
> **Response to Reviewer YaiK**
>
> We appreciate that you find our method to be simple and elegant. We have now made several improvements to the paper in response to reviewers comments as highlighted in the response to all reviewers.
>
> Regarding your feedback specifically:
> * We used the definition of sumnorm that you mention, and added it in line 414 of the revised paper.
> * $\varepsilon$ is a small constant that we set to 1e-6 in our experiments to avoid potential numerical instabilities. We added this in line 418 of the revised paper.

---

### Official Review · Reviewer_NEVA · 2025-11-01

**Soundness:** 3
**Presentation:** 3
**Contribution:** 3
**Rating:** 6
**Confidence:** 4

**Summary:**

This paper introduces DotMatch, a novel semi-supervised learning (SSL) method that omits hyperparameters, such as confidence thresholds or regularization strengths. The key contribution is the Log Dot Product (LDP) loss, a new consistency loss that replaces the traditional cross-entropy loss for learning with unlabeled data. The LDP loss simply computes the log negative dot product between predictions between weak- and strong-augmentations. Moreover, when combined with distribution alignment, DotMatch achieves competitive or superior performance to other baseline methods. Furthermore, a theoretical gradient analysis is provided to support the empirical findings.

**Strengths:**

- This paper proposed a simple and intuitive method named DotMatch that omits hyperparameter tuning, which is easy to employ and simplifies the training process.
- This paper provides a rigorous theoretical framework to justify its method.
- The experiments are quite extensive and show promising performance improvement.

**Weaknesses:**

- While the LDP loss is formulated in a simple and elegant manner, it remains similar to standard consistency-based SSL approaches, where they simply use the prediction consistency between weak and strong augmentation to conduct SSL. As a result, consistency-based SSL either does not require hyperparameters to control the strength or threshold. Could you further justify why the DOT multiplication is different from the MSE or KLD-based consistency training loss?
- Semi-Supervised Learning under open world and distribution shift has been an important research topic that scales SSL from traditional IID scenario to more complicated real-world applications [1], [2], and [3]. How the proposed DotMatch can be successfully employed in open world SSL is worth further discussing and investigating because there is no justification nor experiments to demonstrate the robustness.
[1] Saito et al., Openmatch: Open-set semi-supervised learning with open-set consistency regularization, in NeurIPS 2021.
[2] Huang et al., Universal semi-supervised learning, in NeurIPS 2021.
[3] Yu et al., Multi-task curriculum framework for open-set semi-supervised learning, in ECCV 2020.

**Questions:**

- How does LDP behave when the prediction distributions are nearly orthogonal (dot product close to zero)? Is there a risk of instability in the gradient magnitude?
- Can LDP be extended to multiview or multimodal SSL cases where prediction distributions may differ structurally? Does the proposed method remain effective if augmentations are weaker, i.e., the difference between weak and strong augmentations are small? Will LDP still be effective if the application is related to a different modality, such as Speech or Text SSL?
- Would introducing an adaptive temperature in LDP improve generalization further? How can such a temperature affect the learning performance?

---

> ### Author Response · Authors · 2025-11-26
> **Response to Reviewer NEVA**
>
> We appreciate that you find DotMatch to be simple and intuitive, highlight the rigor in our work and the extensive experiments.
>
> **Comparing LDP, MSE and KLD as consistency losses:**
> We omitted a comparison with the KLD loss in the paper because, when stopping the gradient through the target, KLD and CE(soft) are equivalent. This means they share the exact same gradient and optimization properties. MSE behaves similarly as KLD and CE(soft) in that they share the same minimizer, the soft target distribution. Therefore, without the use of target sharpening or one-hot pseudo-labels, using KLD or MSE will not encourage the model to produce confident predictions on unlabeled examples and lead to worse performance.
>
> **Other problem settings:**
> SSL with open-world assumption, distribution shift and multi-modality are interesting problem settings, but are outside the scope of our work on the standard SSL problem. We chose to focus on image data as this is a standard benchmark used in related works on SSL. In principle, with appropriate data augmentation techniques, applying DotMatch to other modalities such as speech or text is possible.
>
> **Nearly orthogonal predictions:**
> We mention this special case in lines 284-287 when discussing Table 1. The prediction distributions are close to orthogonal when they both have high confidence but for different classes. In this scenario, the implicit learning rate scaler for the LDP loss (Table 1 bottom right) is largest, meaning that the algorithm implicitly assigns a large learning rate for this example to quickly align the predicted distribution with the target.
>
> We did not run into any numerical instability issues in our testing. This is likely because the model naturally seldom predicts a weak and strong augmented version of the same input into different classes with high confidence, and so the dot product will not get small enough to cause issues.
>
> **Weaker augmentations:**
> While we did not explore how the quality of data augmentations affect the performance of DotMatch, a thorough investigation into the effect of data augmentation on performance of consistency training in SSL was performed in [1]. The authors of this paper concluded that higher quality augmentations generally lead to improved performance under the same consistency training framework. This was tested in both language and vision tasks.
>
> [1] Xie et al., Unsupervised Data Augmentation for Consistency Training, NeurIPS, 2020.
>
>
> **Adaptive temperature:**
> The gradients of the LDP loss are already naturally scaled appropriately in terms of the confidence of predictions as we show in the paper. We also emphasized this further in the new analysis provided in the updated version of the paper (Theorem 2, Figure 3). However temperature scaling could, in principle, still be applied to the targets in our consistency objective based on LDP loss.
>
> If targets are sharpened then training will speed up (larger gradients), but at the expense of potentially worsening the confirmation bias problem as low-confidence incorrect pseudo-labels will have their confidence levels increased.
>
> If targets are made more uniform then training will slow down (smaller gradients). This may help with the confirmation bias problem by reducing confidence for incorrect examples, but will also reduce the learning rate for correct examples. The model will therefore be more susceptible to getting stuck in poor local minima.

---

### Official Review · Reviewer_SFsj · 2025-11-02

**Soundness:** 3
**Presentation:** 3
**Contribution:** 3
**Rating:** 6
**Confidence:** 4

**Summary:**

The paper proposes DotMatch, an SSL method that replaces the usual unlabeled-data cross-entropy with a Log Dot Product (LDP) loss. Despite its simple form, LDP (i) automatically down-weights low-confidence unlabeled examples, (ii) implicitly encourages low-entropy predictions, and (iii) matches the gradient scale of supervised cross-entropy—without introducing additional algorithm-specific hyperparameters common in prior work. Experiments show that the proposed method works well on several benchmarks.

**Strengths:**

The paper is clearly written and easy to follow.
The LDP loss naturally (i) down-weights low-confidence unlabeled samples, (ii) encourages low-entropy predictions, and (iii) matches supervised CE’s gradient scale. This hyperparameter-light design could be practically appealing as it minimizes algorithm-specific hyperparameter tuning that many SSL methods require.

**Weaknesses:**

The overall framework closely resembles FixMatch (adding DA), but with the consistency loss replaced by the proposed LDP objective. However, since similar dot-product–based losses have appeared in prior work and are here adapted to SSL, the contribution feels incrementally novel.

**Questions:**

NA

---

> ### Author Response · Authors · 2025-11-26
> **Response to Reviewer SFsj**
>
> We appreciate that you found our paper to be clearly written and easy to follow, and highlight strengths of our method like hyperparameter-light design.
>
> **Novelty:**
> We kindly refer you to the novelty section of the response to all reviewers.

---

### Official Review · Reviewer_MU44 · 2025-11-04

**Soundness:** 3
**Presentation:** 4
**Contribution:** 2
**Rating:** 4
**Confidence:** 4

**Summary:**

This paper introduces DotMatch, a semi-supervised learning (SSL) algorithm centered on a novel Log Dot Product (LDP) loss. The paper's primary and most significant contribution is its theoretical analysis of this loss function. Theorem 1 proves that the LDP loss's gradient norm is implicitly coupled with the target's confidence (entropy): high-entropy (uncertain) targets naturally produce near-zero gradients, while low-entropy (confident) targets produce large gradients. This presents an elegant, implicit mechanism for balancing signal-to-noise, contrasting with the explicit thresholding or re-weighting mechanisms of algorithms like FixMatch and SoftMatch. Empirically, DotMatch shows very strong performance in extremely low-label regimes.

**Strengths:**

## Reasons to Accept

---

* **Novel Theoretical Contribution:** The paper's core strength is its theoretical analysis of the LDP loss's gradient properties. Theorem 1 provides a "first-principles" explanation for an implicit, confidence-based filtering mechanism that emerges from the loss function's geometry alone. This is a novel and elegant contribution to the field.
* **Good Low-Label Performance:** DotMatch achieves state-of-the-art results in the most challenging, data-starved settings, such as EMNIST with 47 labels (~1 per class) and CIFAR-100 with 400 labels.
* **Elegant Loss Formulation:** The LDP loss itself is a clever, unified objective. As shown in the analysis and ablations, it successfully integrates three key SSL goals into one function: consistency regularization (via weak/strong augmentation), inherent entropy minimization (driving predictions toward one-hot targets), and implicit confidence-based re-weighting.
* **Strong Ablation Study (for LDP):** The ablation in Table 3 provides causal evidence for the LDP loss's effectiveness. It shows that LDP+DA (DotMatch) is dramatically better than using a standard Cross-Entropy loss with either hard or soft targets, confirming the LDP loss is the primary driver of the algorithm's success.

**Weaknesses:**

## Reasons to Reject

---

* **Hyperparameters:** The paper's main claim of having "no algorithm specific hyperparameters" is false. The DotMatch objective (Eq 5) explicitly includes a Distribution Alignment (DA) component. As defined in Section 4.3, this DA mechanism depends on $\hat{\pi}_{t}$, an "EMA of the prior," which is calculated using an "EMA momentum $m$". $m$ is algorithm-specific hyperparameter that is left un-ablated.
* **Missing Large-Scale Benchmarks:** The paper's empirical validation is only on small, "classic" datasets (e.g., CIFAR, EMNIST). It lacks experiments on standard large-scale benchmarks like ImageNet or WebVision. Prior work like SoftMatch/FixMatch/FreeMatch are evaluated on these.
* **Empirical Strength:** DotMatch is not universally better. It is significantly outperformed by FixMatch on SVHN (40 labels) and outperformed by FreeMatch and SoftMatch on CIFAR-10 (40 and 250 labels). This suggests its filtering may be an overly conservative liability on "easier" datasets where greedy, explicit methods (like FixMatch) are superior.

**Questions:**

## Questions for the Authors

---

* The paper’s core premise is "no algorithm specific hyperparameters". Why is $m$ not considered an algorithm-specific hyperparameter, and can you provide a full ablation study for it across datasets?
* Why were experiments on standard, large-scale benchmarks like ImageNet and WebVision omitted? SoftMatch and FreeMatch use these to prove scalability and noise robustness. Without them, it is difficult to assess the practical value of DotMatch. Could you add these results?
* Table 2 shows that DotMatch is outperformed by FixMatch on SVHN and by FreeMatch on CIFAR-10. Does this suggest that the LDP loss’s implicit, gradual filtering is actually a disadvantage on datasets where a model can quickly become confident, and that a "greedy" explicit threshold (like FixMatch's) is superior in those cases?

---

> ### Author Response · Authors · 2025-11-26
> **Response to Reviewer MU44**
>
> We appreciate that you find our theoretical contributions to be novel and elegant, and our loss design to be clever.
>
> **EMA momentum m:**
> Since DA and model EMA are standard techniques used in several SSL algorithms, and we did not introduce them as novel components of DotMatch, we do not count m as being an algorithm-specific hyperparameter. Furthermore, as is standard practice, we simply fixed m to be 0.999 for all instances where EMA is used without doing any tuning.
>
> **Missing benchmarks:**
> After carefully checking the original papers again, none of FixMatch, SoftMatch or FreeMatch were evaluated on WebVision as you claim. Furthermore, the most recent baseline in our comparison, InterLUDE (ICML 2024), also does not include any experiments with ImageNet or WebVision.
>
> Regarding ImageNet specifically, we kindly refer you to the ImageNet section in the response to all reviewers.
>
> **DotMatch is not universally better:**
> We would like to point out that in the 3 specific cases you highlight, DotMatch is not significantly outperformed by any method for 2 of these (SVHN with 40 labels, CIFAR10 with 250 labels), as we show by the overlapping confidence intervals in Table 2.
>
> In general, other SSL methods achieve strong performance without being universally better (e.g., the most recent baseline InterLUDE).
>
> As for the potential explanation that DotMatch struggles with "easier" datasets, our results do not support this since MNIST and EMNIST are arguably "easy" in the sense that they contain black and white images, but DotMatch achieves SOTA on these problems.

---

### Official Review · Reviewer_aq1g · 2025-11-05

**Soundness:** 3
**Presentation:** 2
**Contribution:** 2
**Rating:** 4
**Confidence:** 4

**Summary:**

This paper focuses on semi-supervised learning (SSL) and proposes DotMatch, an algorithm that leverages multi-view consistency and distribution alignment (DA) to learn from unlabeled data without algorithm-specific hyperparameters: it first introduces the Log Dot Product (LDP) loss to measure consistency between weakly and strongly augmented views of unlabeled examples, then combines it with distribution alignment to match the predicted label distribution of unlabeled data with that of labeled data; with LDP loss down-weighting low-confidence examples and implicitly minimizing entropy, the framework achieves competitive performance on standard SSL benchmarks. While it has strengths in desirable qualities of SSL, hyperparameter-free design, theoretically grounded gradient analysis, it also faces issues like reliance on lack of novelty, insufficient experimental analysis and presentation issues.

**Strengths:**

1.Desirable qualities on SSL: The method proposes LDP loss with distribution alignment and achieves desirable qualities on SSL, namely consistency, entropy minimization and small gradient norms for low-confidence unlabeled examples.
2.Hyperparameter-free design: The method is hyperparameter-free without needing to tune any algorithm-specific hyperparameters for different datasets.
3.Theoretically grounded gradient analysis: The paper gives theoretically grounded gradient analysis and comparison of three losses (CE(hard), CE(soft) and LDP) through formula derivation and visualization.

**Weaknesses:**

1.Lack of novelty: The proposed method is largely built upon established SSL paradigms. Similar forms of Log Dot Product–based losses have been discussed in prior works, and the use of distribution alignment is also a well-known strategy.

2.Insufficient experimental analysis: The experiments mainly report classification test errors without deeper quantitative or qualitative analyses to support the claimed advantages, such as confidence calibration, the contribution of unlabeled samples, or training dynamics visualization. More ablation or interpretability studies would strengthen the empirical validation.

3.Presentation issues: The paper suffers from inconsistent and potentially confusing notation—such as using both bold and non-bold versions of the same symbol to denote different quantities—and inconsistent symbol definitions.

**Questions:**

1.LDP loss Innovation: The proposed LDP loss shares a similar structural form with the Pairwise Objective introduced in the following ICLR 2022 paper. A clearer distinction between the two should be articulated.
OPEN-WORLD SEMI-SUPERVISED LEARNING. ICLR 2022
2.DA originality: DA is also a commonly used correction algorithm. It is supposed to illustrate whether it has been improved or innovated, and also explain in detail the relevant details of the DA formula and its various symbols.
3.Limited comparison with existing methods: The comparison set in the experiments appears relatively narrow. Beyond the few mentioned baselines, the paper should consider including a broader range of semi-supervised learning methods, particularly recent state-of-the-art algorithms.
4.Dataset-specific performance concerns: The proposed method exhibits significant performance gains primarily on the EMNIST dataset, while the improvements on other benchmarks are marginal. This raises concerns about potential dataset-specific tuning or overfitting.

---

> ### Author Response · Authors · 2025-11-26
> **Response to Reviewer aq1g**
>
> We are glad that the main strengths of DotMatch were conveyed clearly such as achieving desirable qualities of an SSL algorithm and hyperparameter free design.
>
> **Novelty:**
> We kindly refer you to the novelty section of the response to all reviewers.
>
> **Experimental analysis:**
> We kindly refer you to the empirical and theoretical contributions section of the response to all reviewers, particularly C2.
>
> **Presentation:**
> We have carefully checked the paper again and do not see any instance of using both bold and non-bold versions of the same symbol to denote different quantities.
>
> We are happy to address any notational issue if the reviewer could provide specific details.
>
> **Distribution alignment:**
> We added missing details about the DA formula and its symbols to the updated version of the paper, including the definition of sumnorm on line 414, $\varepsilon$ on line 418, and element-wise product $\odot$ in the caption of Table 1 where it first appears.
>
> **Limited comparison with existing methods:**
> We included FixMatch as a simple but strong baseline, SoftMatch and FreeMatch because of their strong performance and general association with being SOTA in this problem setting, and FlatMatch and InterLUDE as recent SOTA methods that were released after both SoftMatch and FreeMatch. These baselines are also highly relevant to DotMatch in the sense that they are all single-model methods that do not utilize any auxiliary models other than the main classifier during training. The baselines we chose have already been shown to outperform several other SSL methods in their original papers, so we did not feel the need to include each of those in our experiments.
>
> **Concerns about potential dataset-specific tuning or overfitting:**
> As we highlighted in the response to all reviewers, one of the salient features of DotMatch is that it removes common algorithm-specific hyperparameters in SSL such as confidence threshold and regularization constants, allowing it to perform well on new problems (like EMNIST) without requiring tuning of these values. Furthermore, as is shown in our results tables, DotMatch also performs very well on datasets other than EMNIST.
>
> Regarding algorithm-independent hyperparameters used for DotMatch (learning rate, weight decay, momentum, etc.), these values were not tuned, instead, we simply used standard values from the USB benchmark.

---

### Official Review · Reviewer_q8in · 2025-11-07

**Soundness:** 2
**Presentation:** 2
**Contribution:** 2
**Rating:** 2
**Confidence:** 3

**Summary:**

This paper proposes the DotMatch method for semi-supervised learning (SSL) problem, with the proposed log dot product loss (LDP) loss applying to a classic form of SSL objective functions. Through optimum and gradient analysis of LDP, the authors claim that LDP has benefits that non-confident examples have a low contribution to the gradient and that the optima are encouraged to have a low entropy. These properties of LDP enable the proposed DotMatch method to be free of hyperparameter tuning. Experimental comparisons with other SSL methods on classic CV benchmarks are also conducted.

**Strengths:**

- The proposed method requires no algorithm-specific hyperparameters.
- The proposed method is easy to understand and implement.
- The proposed LDP loss is analyzed from both theoretical and experimental perspectives.

**Weaknesses:**

- The experimental results are weak. 1) In Table 2, the proposed DotMatch method achieves the best accuracy in only 5 out of 10 cases. 2) The experiments are conducted on relatively small scale CV benchmark datasets. Results on larger datasets such as Imagenet are lacking.
- As shown in Figure 2, when the target is close to uniform, LDP requires far more gradient steps than CE to reach the optimum. This raises my concerns about the computational efficiency of DotMatch. To address this, I think experimental comparisons on the training time of LDP with other methods should be reported.
- Figure 1 should include DotMatch to better support the claims.
- In Table 1, the notations $\boldsymbol{z}$ and $p \odot q$ are used but undefined.
- Line 930. Table 7 should be Table 2?

**Questions:**

- I cannot find which section supports the claim in the abstract that "its gradient is appropriately scaled relative to the gradient of the supervised loss, thus requiring no regularization constant." Could you specify this for me?

---

> ### Author Response · Authors · 2025-11-26
> **Response to Reviewer q8in**
>
> Thank you for acknowledging several strengths of our work, including no algorithm-specific hyperparameters,
> easy to understand and implement, and LDP loss analyzed from both theoretical and experimental perspectives.
>
> **Weak experimental results:**
> We kindly refer you to the empirical and theoretical contributions section of the response to all reviewers, particularly C1, as well as the ImageNet section.
>
> **Computational efficiency:**
> Please refer to C3 in the response to all reviewers.
>
> Note that Figure 2 does not imply that DotMatch is computationally inefficient, but shows DotMatch can effectively mitigate the influence of unconfident examples, while for algorithms using CE loss, unconfident examples can still contribute to large gradients.
>
> **Figure 1:**
> We have added DotMatch to Figure 1, which shows that similarly to SoftMatch and as expected, it down-weights the gradient norm of low-confidence examples compared to CE(hard), thus leading to high quality gradients and more accurate final model.
>
>
> **Undefined notations:**
> We have clarified this in Table 1 caption in the revised version of the paper.
>
> **Line 930:**
> Yes, Table 7 should have been Table 2. Fixed.
>
>
> **Gradient scaling:**
> We have clarified this by adding Figure 3, Theorem 2 and a discussion at the end of Section 4.2. From the gradient formulas of the different losses in Table 1, we can see that no matter whether the predicted class distribution is certain or not, the standard cross entropy loss will treat the pseudo-label equally as the supervised loss; in contrast, the gradient of the LDP loss is implicitly scaled in a way such that it is closer to 0 when the predicted distribution is more uncertain. Theorem 2 suggests that the gradient scaling factor naturally decreases to 0 as the uncertainty of the target distribution increases. Such scaling is demonstrated in Figure 2 and Figure 3, and its effectiveness is supported by the competitive performance of DotMatch against the baselines.

---

### Author Response · Authors · 2025-11-26
**Response to all Reviewers**

We thank the reviewers for their positive comments such as our algorithm DotMatch being easy to understand and achieving desirable qualities of a semi-supervised learning algorithm (consistency, entropy minimization, small contributions to learning from unconfident unlabeled examples), and highlighting our theoretical analysis of loss gradients.

We have also revised our paper significantly with new empirical and theoretical results and improved writing to address comments from the reviewers.

We highlight and clarify the novelty, and empirical and theoretical contributions of our work below:

**Novelty:**
DotMatch is empirically competitive with state-of-the-art (SOTA) algorithms, but distinguishes itself from other algorithms by its simple and algorithm-specific hyperparameter-free design, grounded on novel empirical and theoretical analysis. Our empirical and theoretical analysis shows that DotMatch has SSL properties such as consistency, entropy minimization, and a gradient scaling behavior that down-weight the gradient of uncertain examples in an elegant way. These have been further supported with new empirical and theoretical results, and are detailed in the Contributions section below

While similarly to previous algorithms like SoftMatch and FreeMatch, our algorithm builds on existing general ideas like consistency regularization and distribution alignment, we believe that the above features distinguish our work from existing ones and these novel features are significant.

Our independently developed LDP loss resembles the pairwise objective in [1], but unlike [1], we provide extensive empirical and theoretical analysis. In addition, LDP is designed to enforce confident and consistent predictions across augmented views of unlabeled inputs, while the pairwise objective targets novel-class discovery and pseudo-labels unlabeled data using nearby labeled examples. We have added [1] to the related work section.

**Contributions:**
We have added new empirical (computational efficiency comparison, further empirical analysis on gradient scaling behavior of the LDP loss) and theoretical (Theorem 2 on LDP loss gradient scale)
results  to the paper. We believe these new results (**bolded** below), together with the original ones, are substantial and provide insights and support on DotMatch's strong performance across datasets of diverse scale and complexity.

C1. We demonstrate DotMatch's strong performance (without hyperparameter tuning) across datasets with 10 to 100 classes and 50 000 to 604 388 training examples.
Table 2 shows that DotMatch achieves the highest average best test accuracy for 5 out of 15 settings (more than all baseline methods), and it is not significantly outperformed in other settings.
In addition, Table 7 shows that DotMatch is generally competitive or better than the baseline methods in terms of average final test accuracy.

C2.
We provide empirical and theoretical support for the advantages of DotMatch:

1. Low-confidence examples naturally contribute less to the gradient update without requiring additional confidence-based loss weights. See **revised Figure 1**, Theorem 1, **Theorem 2**, and visual comparisons of the dynamics of gradient descent in Figure 2.
2. The model will automatically learn to produce high confidence predictions, justified by examining the properties of the loss minimizer in line 260.
3. LDP gradients are appropriately scaled relative to the gradients of the supervised CE loss: we have clarified this by adding **Theorem 2**, **Figure 3**, and a discussion at the end of Section 4.2.

C3.
We demonstrate that DotMatch is computationally efficient by comparing its per iteration run time with baseline methods in **Table 8**. The results show that DotMatch is consistently the fastest algorithm in our comparison.


**ImageNet:**
As suggested by some reviewers, to further expand the range of dataset scales in our comparison, we have started running experiments on ImageNet. However, based on our current progress, the running time will be unfortunately long: running each algorithm once on ImageNet using an NVIDIA H100 GPU is estimated to take 18 days, and the estimated total running time for performing the same benchmarking experiments as on other datasets is $18 \times 6 \times 3 = 324$ days.

[1] Cao et al., Open-World Semi-Supervised Learning, ICLR, 2022.

---

### Meta-Review · Area_Chair_Kzki · 2026-01-01

**Summary:**

The primary reason for the recommendation to reject is the lack of empirical evidence supporting the scalability and universal superiority of the proposed DotMatch algorithm.

Specifically, three reviewers (q8in, aq1g, MU44) raised significant concerns regarding the absence of large-scale experiments (ImageNet/WebVision). The authors' rebuttal admitted that such experiments were not completed, and their estimated timeframe (324 days) suggests a lack of preparedness for the current competitive landscape of SSL research. Furthermore, the core claim of being "algorithm-specific hyperparameter-free" was successfully challenged by Reviewer MU44, who pointed out that the EMA momentum ($m$) remains a tunable parameter, undermining the paper's primary conceptual contribution. Finally, the empirical results on smaller datasets were inconsistent, with DotMatch often being outperformed by older baselines like FixMatch and SoftMatch on standard benchmarks like SVHN and CIFAR-10.

**Reviewer Concerns:**

**Addressed by the Rebuttal:**
*    The authors clarified the missing definitions for `sumnorm`, $\epsilon$, and other mathematical symbols (Addressing concerns from YaiK, aq1g, and q8in).
*    The authors provided per-iteration runtime comparisons (Table 8) showing that DotMatch is computationally efficient on a per-step basis (Addressing q8in).
*    The addition of Theorem 2 and Figure 3 helped explain the gradient scaling properties of the LDP loss compared to MSE/KLD (Addressing NEVA).

**Outstanding Concerns (Reasons for Rejection):**
*   The most critical deficiency is the lack of ImageNet results. In modern SSL research, demonstrating that a method works on small-scale datasets (CIFAR/EMNIST) is insufficient, especially when the authors argue their method eliminates the need for the tedious tuning that usually occurs at scale.
*   The authors’ defense that the EMA momentum $m$ is "standard" does not negate the fact that it is an algorithm-specific parameter. Reviewers (MU44) remained unconvinced that DotMatch truly removes the need for hyperparameter selection.
*    As noted by q8in and MU44, the method is not "universally better." It achieves SOTA only in a subset of low-label regimes and struggles on "easier" datasets compared to existing greedy methods. This suggests the LDP loss may have a "conservative" filtering bias that limits its practical utility across diverse data distributions.

**Reviewer Scores:**

*   **Reviewer q8in (Initial: 2):** **Remains 2.**
*   **Reviewer aq1g (Initial: 4):** **Remains 4.**
*   **Reviewer MU44 (Initial: 4):** **Remains 4.**
*   **Reviewer SFsj (Initial: 6):** **Remains 6.**
*   **Reviewer NEVA (Initial: 6):** **Remains 6.**
*   **Reviewer YaiK (Initial: 6):** **Remains 6.**

---

### Decision · Program_Chairs · 2026-01-26

Reject